# Enzyme-inspired single-atom photocatalysis for oxygen reduction to hydrogen peroxide

Lukáš Zdražil [1,2,3], Alejandro Cadranel [1,4,5] ✉, Giorgio Zoppellaro [2,3], Ayse Günay-Gürer[1], Zdeněk Baďura[2,3], David Panáček[2,3,6], Hana Kmentová[2,3], Camila Otero[7], Štěpán Kment [2,3], Maria Ana Huergo[7], Emiliano Fonda [8], Radek Zbořil [2,3] ✉ & Dirk M. Guldi [1] ✉

Photocatalysis presents a promising route for generating sustainable, high-energy-density fuels. However, conventional photocatalysts based on rigid binary metal compounds face significant limitations, including fixed band gaps, rapid charge recombination, and non-specific reaction pathways - ultimately leading to limited selectivity and yield. Critically, they lack the site-specific selectivity characteristic of enzymatic systems, a feature essential for achieving high efficiency, control, and precision. Inspired by cytochrome $c$ oxidase, we report the development of Cu-single-atom-enhanced carbon dots as the enzymatic-like photocatalyst. By mimicking enzyme´s site-specific electron transfer cascade, these carbon dots enable the selective photocatalytic reduction of oxygen to hydrogen peroxide under ambient conditions. This study introduces a strategy for translating enzymatic precision into photocatalytic materials design, bridging molecular and materials catalysis for sustainable energy and chemical transformations.

Renewable energy sources like solar and wind are essential alternatives to fossil fuels. Yet, their intermittent nature presents significant challenges for the seamless integration into energy systems[1]. Artificial photosynthesis offers a promising solution by converting sunlight into portable and storable, high-energy-density fuels[2]. Its widespread implementation is, however, hindered by the limitations of current photocatalysts, which are predominantly based on metal binary compounds that have fixed band gaps[3,4]. Strategies like defect engineering and elemental doping aim at improving catalytic efficiency, but remain impractical for scalable applications[5]. Central to artificial photosynthesis is the two-electron photo-reduction of oxygen to hydrogen peroxide ($H_2O_2$) having immense potential as a high-energy oxidant for applications across medicine, industry, and environmental management[6]. Recent advancements in $H_2O_2$ photo-production technologies have highlighted promising earth-abundant photocatalysts, including transition metal oxides[7,8], noble metal-supported photocatalysts[9–11], covalent organic frameworks[12,13], conjugated polymers[14,15], carbon nitrides[16–19], and organic aerogels[20]. Despite these developments, significant challenges such as rapid charge carrier recombination and undesired photo-induced pathways constrain their practical applicability and limit their full industrial potential in sustainable energy systems[21].

[1]Department of Chemistry and Pharmacy & Interdisciplinary Center for Molecular Materials (ICMM), Physical Chemistry I, Friedrich-Alexander-Universität Erlangen-Nürnberg, Egerlandstraße 3, Erlangen, Germany. [2]Nanotechnology Centre, Centre for Energy and Environmental Technologies, VSB – Technical University of Ostrava, 17. listopadu 2172/15, Ostrava-Poruba, Czech Republic. [3]Regional Center of Advanced Technologies and Materials, The Czech Advanced Technology and Research Institute (CATRIN), Palacký University Olomouc, Šlechtitelů 27, Olomouc, Czech Republic. [4]Universidad de Buenos Aires, Facultad de Ciencias Exactas y Naturales, Departamento de Química Inorgánica, Analítica y Química Física, Pabellón 2, Ciudad Universitaria, Buenos Aires, Argentina. [5]CONICET – Universidad de Buenos Aires, Instituto de Química-Física de Materiales, Medio Ambiente y Energía (INQUIMAE), Pabellón 2, Ciudad Universitaria, Buenos Aires, Argentina. [6]Center for Advanced Technologies and Engineering (CATEN), Technologická 375/3, 708 00, Ostrava-Pustkovec, Czech Republic. [7]Instituto de Investigaciones Fisicoquímicas Teóricas y Aplicadas (INIFTA), Universidad Nacional de La Plata, CONICET, La Plata, Argentina. [8]Synchrotron SOLEIL, Départementale 128, Saint-Aubin, France. ✉e-mail: ale.cadranel@fau.de; radek.zboril@upol.cz; dirk.guldi@fau.de

Addressing these limitations requires a conceptually new approach based on structurally tunable photocatalysts that are capable to trigger selective electron cascades thus mimicking precise principles known from enzymatic catalysis. In nature, reductive enzymes such as cytochrome $c$ oxidase (C$c$O) use metal ions at their active sites to achieve highly selective reduction processes[22]. It operates through a sophisticated electron transfer pathway, starting with cytochrome $c$ as the initial electron reservoir and involving multiple redox centers, including CuA, heme $a$, and the heme $a_3$/CuB binuclear center (BNC). Hereby, electrons flow from cytochrome $c$ to CuA, then to heme $a$, and finally to BNC, where molecular oxygen is reduced to water in a tightly regulated process[23]. Mimicking C$c$O's reduction of molecular oxygen represents a huge challenge mainly due to the above-mentioned limitations of current photocatalysts and nanomaterials possessing very limited structural tunability.

To address this challenge, we decided to combine structural versatility of carbon dots (CDs) with single-atom engineering enabling the design of catalytic sites that replicate the precision and efficiency of enzymatic systems. CDs, unlike traditional binary metal based semiconductors, exhibit tunable photocatalytic and structural properties, positioning them as ideal candidates for next-generation photocatalysts[24]. CDs ability to generate long-lived spin-separated species enhances charge separation efficiency, which is a critical factor for improving photocatalysis[25]. Moreover, CDs showed photocatalytic potential in oxygen reduction reaction (ORR) and water oxidation reaction (WOR)[25–27].

In this study, we demonstrate that single-atom-engineered CDs effectively mimic the function of C$c$O. Under photo-excitation, the initial reduction is driven by a sacrificial electron donor (triethanolamine, TEOA), followed by a controlled electron transfer from the carbon dot domain (CD) to the copper-containing active site (CuP), where oxygen is bound and reduced. This sequence parallels cytochrome $c$, the CuA site, and the CuB site, respectively. Such a bio-inspired design of CDs powers efficient $H_2O_2$ photo-production. By bridging enzymatic principles with advanced photocatalysis, this approach paves the way for designing new generation of enzyme-like photocatalysts with accurate reaction control.

## Results

The design of C$c$O mimics is based on single-atom engineered CDs. The highly functionalized nature of CDs poses significant challenges for placing single $Cu^{2+}$ ions at specific sites[28]. To overcome this, we focused on CDs featuring porphyrinoids integrated into their structures, labelled as $H_2P@CD$[29]. This strategy leverages the high affinity of tetrapyrroles for metal cations[30]. Thus, the enzyme-like photocatalyst based on CDs was fabricated through a two-step process: synthesis of $H_2P@CD$, followed by coordination of $Cu^{2+}$ (CuP@CD). Transmission electron microscopy (TEM) of the freshly synthesized $H_2P@CD$ revealed particles with an average size of ~2.8 nm (Fig. S1).

### Electronic characterization of $H_2P@CD$ and CuP@CD

The optical properties of $H_2P@CD$ and CuP@CD were analyzed using steady-state and time-resolved optical spectroscopy in diluted aqueous suspensions at room temperature. The absorption of $H_2P@CD$ exhibited distinct features from both the carbon domains and porphyrinoid sites, including Soret-band absorptions at 395 and 417 nm and Q-band absorptions at 575, 600, 625, and 675 nm (Fig. 1a). Excitation-emission color maps revealed, on one hand, the excitation-dependent fluorescence of CDs and, on the other hand, the excitation-independent fluorescence of porphyrinoids[29,31]. In particular, maxima in the range from 450 to 515 nm and maxima at 680 and 650 nm stand out (Fig. 1b, S2a)[32]. From time-resolved emission spectroscopy (TRES) with $H_2P@CD$, we derived CD-centered lifetimes of 0.8 and 5.3 ns, while porphyrinoid-centered lifetimes are 1.7 ns at 650 nm and 5.1 ns at 680 nm (Fig. 1d, f). Considering the aforementioned these three

distinct emissions are assigned to (i) CD-centered emission in the 450–515 nm range, (ii) porphyrinoid emission at 650 nm (P650), and (iii) porphyrinoid emission at 680 nm (P680). Not only that these species differ in terms of their emission maxima, but also in terms of their radiative decays. As such, $H_2P@CD$ is composed of structurally distinct emissive centers. It also confirms the modular nature of $H_2P@CD$ and sets the basis for a dual functionality that mimics the electron reservoir and the copper active sites known for C$c$O.

Upon addition of $Cu^{2+}$, the Soret-band absorptions blue-shifted to 390 and 407 nm, and the four Q-band absorptions coalesced into two at 575 and 635 nm. All these changes indicate differences in the electronic structure of $H_2P@CD$ and CuP@CD (Fig. 1a). A closer look at the fluorescence reveals that $Cu^{2+}$ coordination quenches one of the porphyrinoid domains, while the impact on the CD-centered emission is subtle (Fig. 1c). Among others, $Cu^{2+}$ coordination causes a quenching of the 680 nm fluorescence and lifetime thereof, but not of the 650 nm fluorescence (Fig. 1e, g, and S2b). Only P680 binds $Cu^{2+}$, but not P650. Uniformity of CuP@CD was independently confirmed by size-exclusion column chromatography. Essentially, identical emissions were detected for all fractions (Fig. S2c, d). This rules out that porphyrinoid species are not integrated parts of either $H_2P@CD$ or CuP@CD. Concentration-dependent measurements, during which $[Cu^{2+}]$ was varied, confirmed the selective coordination (Fig. S3). In $H_2P@CD$ and CuP@CD, their spectral features closely resemble the benchmark free-base tetrakis(4-carboxyphenyl)porphyrin ($H_2$TCPP) and its copper complex (CuTCPP) (Fig. S4).

Deeper insight into the excited-state dynamics within the porphyrinoid sites came from femtosecond transient absorption spectroscopy (fsTAS) measurements. These were performed using 420 nm photo-excitation into the Soret-band absorption. Ground-state bleaching (GSB) at 385 – 430 nm and 600 – 700 nm, were accompanied for $H_2P@CD$ by excited-state absorption (ESA) across the visible range. Target analysis revealed three species – with lifetimes of 4 ps, 2 ns, and 5 ns – along with an infinite offset from a long-lived species, reflecting the vibrational relaxation and decay pathways of P650 and P680 (Fig. 1h, S5). On one hand, P680 initially exhibited features stemming from a hot excited singlet state (hot-$S_{680}$), which relaxed within 4 ps to populate the relaxed singlet excited state ($S_{680}$), which decayed with a lifetime of 5 ns. On the other hand, P650 showed a GSB at 645 nm and a singlet excited state ($S_{650}$) decay of 2 ns. Turning to CuP@CD, the P680 decay dynamics were as short as 46 ps, while P650 remained unaffected (Fig. 1i, S6). Such quenching is linked to an enhanced intersystem crossing (ISC) driven by spin-orbit coupling from $Cu^{2+}$ (Fig. S7) to afford the triplet excited states ($T_{680}$ / $T_{650}$). It highlights how $Cu^{2+}$ modulates P680's photophysics and underscores its potential for site-specific reactivity. Photoexcitation of $H_2P@CD$ and CuP@CD at, for example, 350 nm was also explored to investigate the excited-state dynamics upon CD-based excitation[33,34]. The transient spectra are, however, even at early times dominated by those seen when photoexciting the porphyrinic sites at 420 nm excitation (Fig. S8). This is consistent with weak CD-centered transient absorptions, as typically observed in CD-based systems[35–38].

### Structural characterization of CuP@CD

As it is of utmost importance in catalysis, we explored the $Cu^{2+}$ coordination along with the structural features of CuP@CD, in the context of, for example, metal-nitrogen-carbon (MNC) single-atom catalysts. MNC include a porous carbonaceous material with $MN_4$ pyrrolic coordination sites, bearing structural and catalytic similarities with our single-atom engineered CDs. Their performance in ORR[39], (hydrogen evolution reaction) HER[40], and other photocatalytic transformations is excellent[41], albeit they contain several different metal centers[42]. Presence of the later stems from a preparation method that is based on the pyrolysis of different M, N, and C sources. In contrast, our measurements demonstrate that only one copper species is present in

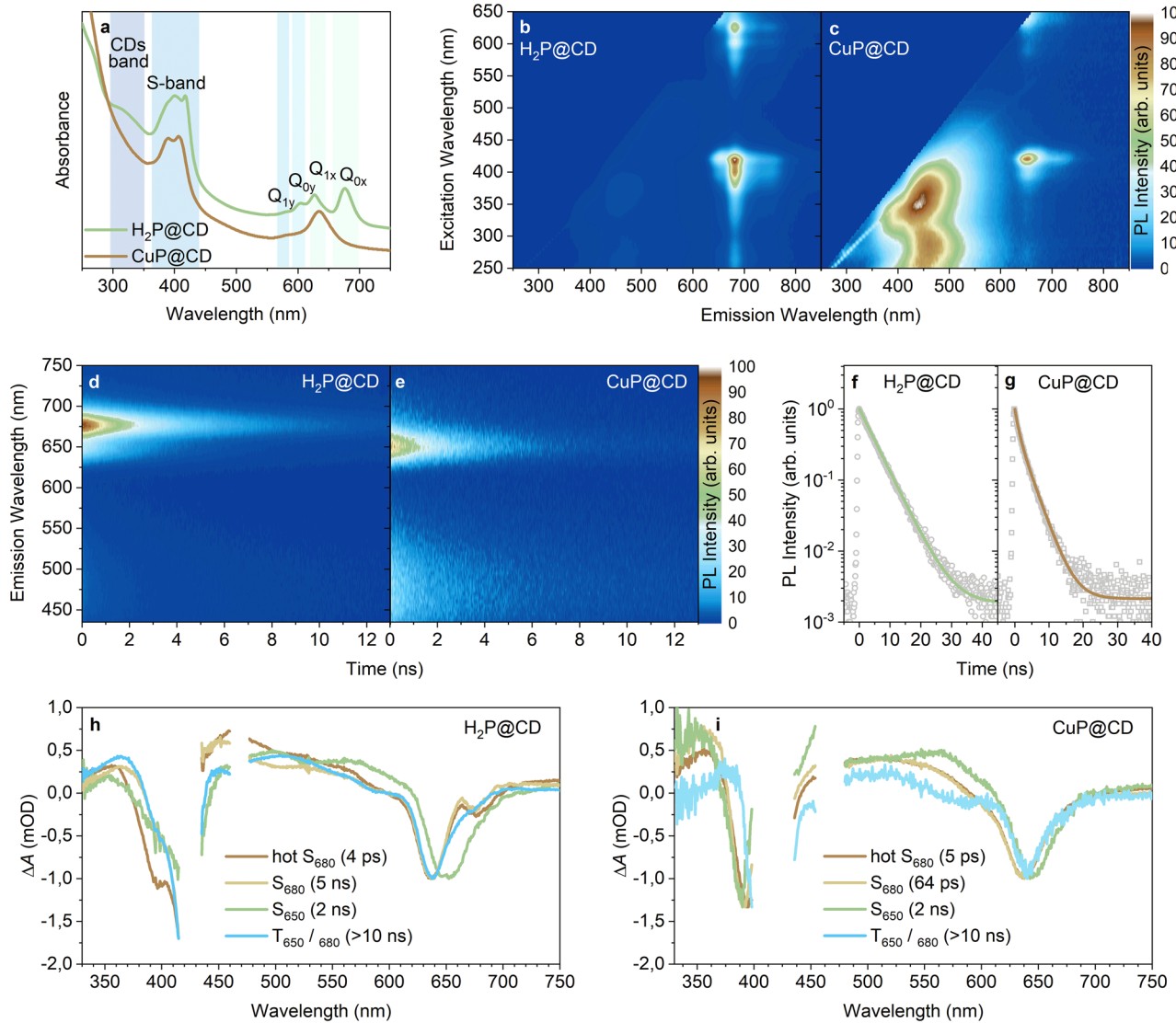

**Fig. 1 | Optical characterization of H₂P@CD and the effect of Cu coordination.** **a** Absorption spectra of H₂P@CD and CuP@CD, highlighting distinct features of the carbon domain and the porhyrinoid sites. **b**, **c** Normalized excitation-emission color maps of H₂P@CD and CuP@CD, illustrating the excitation-dependent CD fluorescence and excitation-independent porphyrinoid fluorescence. **d**, **e** Time-resolved emission color maps of H₂P@CD and CuP@CD ($\lambda_{ex}$ = 355 nm), showing the evolution of fluorescence over time. **f**, **g** Fitting of TRES data ($\lambda_{em}$ = 680 nm and $\lambda_{em}$ = 650 nm) for H₂P@CD and CuP@CD, derived from global analysis. **h**, **i** Species-associated differential spectra of H₂P@CD and CuP@CD derived from fsTAS. $\Delta A$ denotes transient absorbance change, and colors correspond to globally fitted species-associated spectra as indicated. All measurements were conducted on diluted colloidal suspensions at room temperature.

CuP@CD. The average size of CuP@CD remained unchanged upon Cu²⁺ up-take, ruling out metal-induced aggregation (Fig. 2a). X-ray photoelectron spectroscopy (XPS) revealed 2.5 at.% of Cu²⁺ (Fig. S9). Elemental mapping (EDS) showed a homogeneous distribution of Cu²⁺ across the CuP@CDs (Fig. 2b, c; Fig. S10). Any other forms of copper were all ruled out via high-angle annular dark-field scanning transmission electron microscopy (HAADF-STEM) (Fig. 2d), X-ray absorption near-edge structure (XANES), which showed neither the characteristic peak at 8984 eV for Cu⁺ nor any evidence of Cu⁰ at 8979 eV (Fig. 2e), and Cu LMM analysis (Fig. S11). Extended X-ray absorption fine structure (EXAFS) analysis further excluded contributions from Cu⁰ or Cu²⁺ oxides (Fig. 2f; Fig. S12–14 and Table S1). All of this is consistent with an exclusive presence of Cu²⁺ at P680 sites. EXAFS also revealed a higher coordination number of Cu²⁺ in CuP@CD compared to CuTCPP. This suggests that a major fraction in CuP@CD is axially coordinated. This coordination environment is more electron-rich when compared to the coordination in CuTCPP, which leads to a shift toward lower Cu 2p binding energies, as observed in

Fig. 2g. A minor component seems, however, to be identical to the coordination in CuTCPP (Fig. 2g)[43]. Furthermore, a decrease in the pyrrolic N signal and shifts in the N 1s as well as O 1s signals emerges upon Cu²⁺ coordination. We conclude predominant coordination with the nitrogen atoms of the porphyrinoids and axial interactions with oxygen-containing entities (Fig. 2h, i) pendant from the CD domains. Fourier-transform infrared (FT-IR) spectroscopy revealed shifts in the $\nu$(C = O) and $\nu$(C = N) vibrational bands, consistent with interactions between Cu²⁺ and oxygen next to nitrogen-containing entities (Fig. 2j)[31,44]. At acidic conditions (pH 4), the Cu²⁺ EPR envelope is axially symmetric with $g_{\perp}$ (2.36) > $g_{\parallel}$ (2.05) and an effective g-value ($g_{eff}$) of 2.26. This is in support of an in-plane z-y coordination. At neutral pH (pH 7), the system evolved into an isotropic resonance signal with $g_{iso}$ = 2.16 in line with a weakening of the axial field. In contrast, under basic conditions (pH 9), a pronounced rhombic distortion was observed. Such a distortion is consistent with stronger donor interactions that act out of the y-z plane with Cu²⁺[45]. In this scenario, the g-tensor values are $g_z$ = 2.26, $g_y$ = 2.12, and $g_x$ = 2.05 ($g_{eff}$ = 2.18).

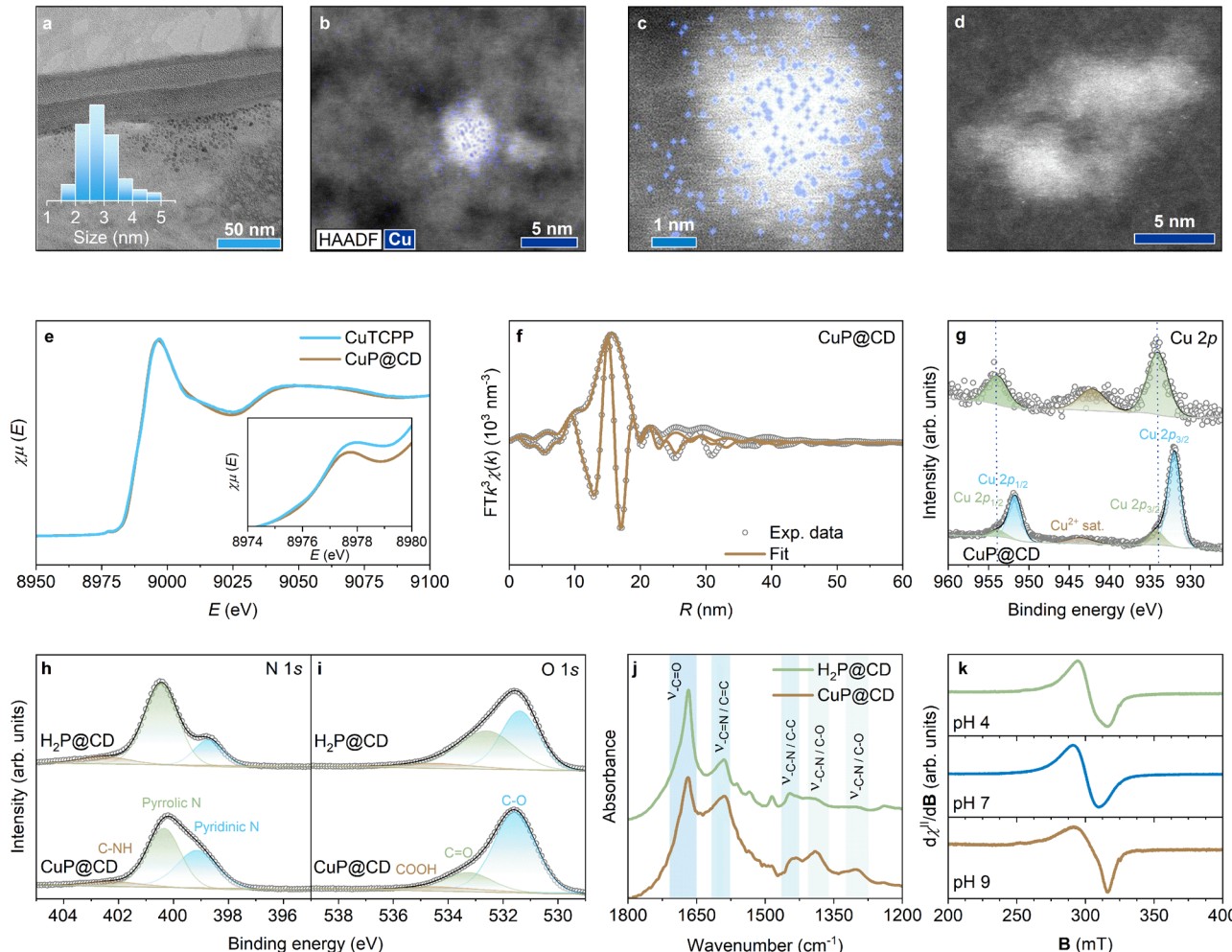

**Fig. 2 | Structural characterization of CuP@CD. a** TEM image of CuP@CD with corresponding size distribution. **b, c** EDS chemical mapping of CuP@CD. **b** shows the high-resolution Cu mapping obtained from HyperMap data using the Automatic Filter mode, while (**c**) shows the corresponding mapping processed with the Smooth Filter (parameter = 3). The inset highlights the uniform distribution of Cu signals across CD, confirming the spatial homogeneity of Cu sites within the CuP@CD ensemble. **d** HAADF-STEM image of two individual CuP@CD. **e** XANES spectra of CuTCPP (light blue) and CuP@CD (brown), with inset showing an enlarged view of the pre-edge region highlighting subtle spectral differences that point toward a more complex coordination environment in CuP@CD. **f** Fourier-transformed (FT) EXAFS data of CuP@CD and its simulation, using a single nitrogen shell as nearest neighbors. **g** High-resolution XPS spectra of Cu 2$p$ for CuTCPP and CuP@CD. **h, i** High-resolution XPS spectra of the N 1$s$ and O 1$s$ regions for H$_2$P@CD and CuP@CD. **j** FT-IR spectra of H$_2$P@CD and CuP@CD. **k** Continuous-wave (CW) X-band (9.07 – 9.08 GHz, $T$ = 80 K) EPR spectra of CuP@CD recorded at different pH values in a frozen aqueous matrix.

Additionally, the $^{63}$Cu nuclear hyperfine component ($I$ = 3/2) is discernible along $g_z$, with $A_z$ = 16 mT. By virtue of an acid-base equilibrium within the CD domains, a penta-coordinative form is favored at basic pH (Fig. 2k).

**Mimicking C$c$O: photo-induced electron cascade in CuP@CD**

The conversion of molecular oxygen requires a sufficient reduction potential[46]. To this end, photo-induced electron transfer of CuP@CD was evaluated using the soluble redox mediator methyl viologen [MV$^{2+}$; $E^{0'}$ = − 0.64 V vs. Ag/AgCl at pH 7]. At acidic pH (pH 5), the photo-activity of CuP@CD was negligible when ascorbic acid (0.2 M) was added as a sacrificial electron donor (SED). The photo-activity increased significantly at neutral pH using ethylenediaminetetraacetic acid (EDTA, 0.1 M) and triethanolamine (TEOA, 10%) and maximized at basic pH (pH 9) using TEOA (Fig. S15a). We take this trend to postulate a favorable band alignment, an efficient electron donation from deprotonated TEOA, and an electron transfer reactivity of photo-excited CuP@CD – all under basic conditions (Fig. S15b, c)[47].

To probe the site-specific reactivity, Stern-Volmer analyses demonstrated that only the CD-sites interact with TEOA, while the porphyrinoid sites (CuP sites) remained inactive (Fig. 3a–c). Thus, TEOA acts as an artificial electron donor, analogous to cytochrome $c$. Indeed, this behavior mirrors the initial electron transfer step in C$c$O, where electrons are funneled from cytochrome $c$ to CuA, and documents the spatial control over the reactivity within CuP@CDs.

Light-induced electron paramagnetic resonance (LEPR) spectroscopy gave insights into the dynamics of the photo-excited species in CuP@CDs. The LEPR signal at $g \approx 2.00$ is attributed to CD-based radicals and an additional distinct feature around 300 mT is characteristic of Cu$^{2+}$[48,49]. Upon 325 nm photo-excitation, the $g \approx 2.00$ signal resolution progressed due to the localization of the light-induced spin-active species localized at the CD sites (Fig. 3d, e)[25]. Under basic conditions (pH 9), a decrease in the $g \approx 2.00$ signal intensity and the simultaneous disappearance of the Cu$^{2+}$ fingerprints around 300 mT suggest the partial photo-reduction of Cu$^{2+}$. It affords Cu$^+$ at the CuP sites and is driven by an electron transfer from CD (Fig. 3f). Computer simulation of the LEPR envelope revealed isotropic features. These correspond to the CD domains and nitrogen hyperfine structures associated with the porphyrinoids and confirm the electron transfer from CD to the CuP site (Fig. 3f–h). The

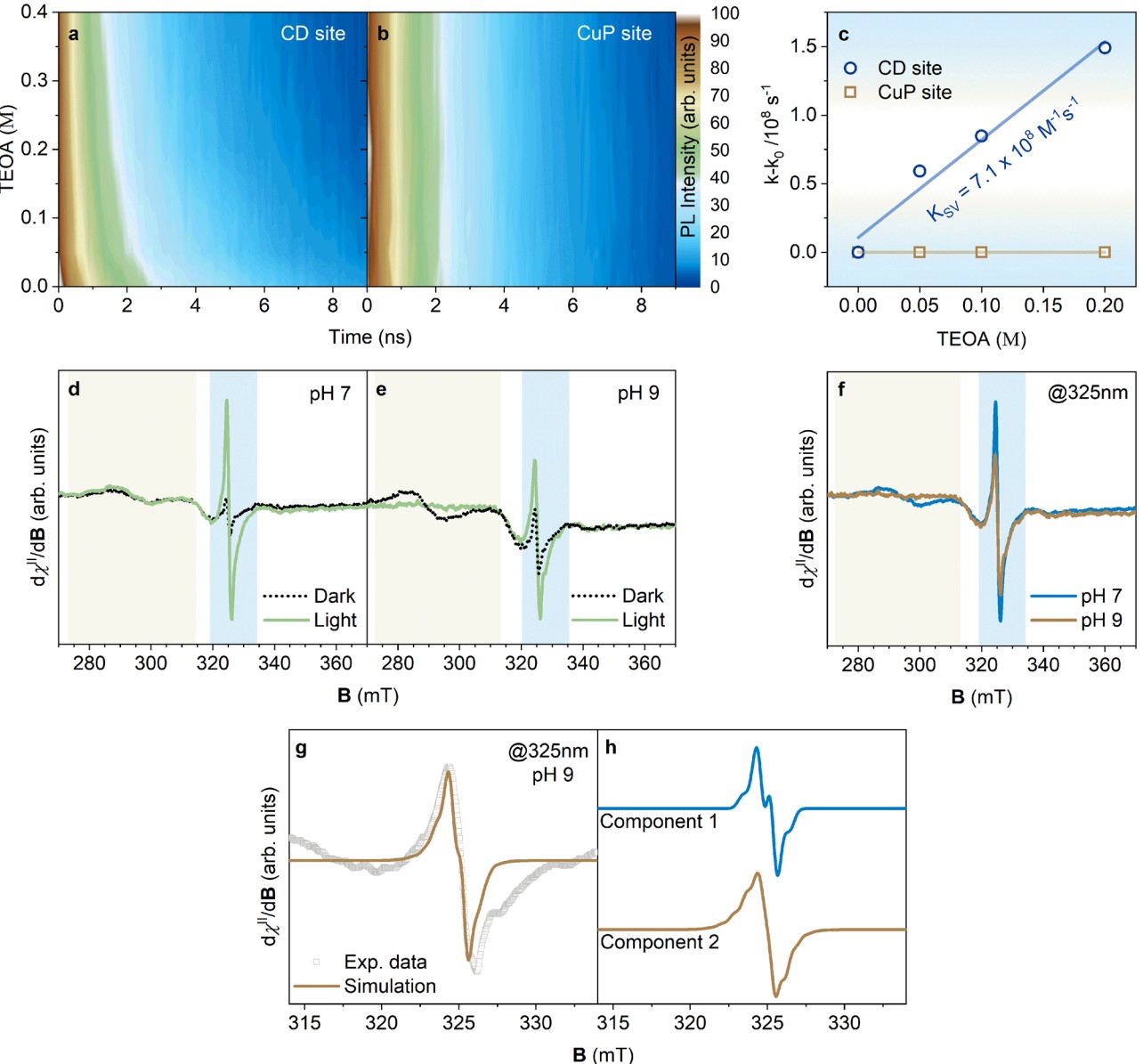

**Fig. 3 | Photoinduced electron-transfer cascade and site-specific charge-separation dynamics in CuP@CD. a**, **b** Color maps of normalized time-resolved fluorescence intensity at the individual active sites ($\lambda_{em} = 450$ nm, $\lambda_{em} = 650$ nm) as a function of TEOA (triethanolamine) concentration (pH 9). **c** Stern–Volmer plots derived from the time-resolved measurements in (**a**, **b**). **d**, **e** X-band (9.07–9.08 GHz) light-induced electron paramagnetic resonance (LEPR) spectra of CuP@CD with 1 ppm of $Cu^{2+}$ at pH 7 and pH 9, respectively. Measurements were conducted in a $N_2$ atmosphere at 90 K with in-situ 325 nm photo-irradiation (40 mW). **f** Comparison of CuP@CD LEPR spectra at pH 7 and pH 9, highlighting an electron transfer at basic pH. **g** Detailed LEPR spectrum of CuP@CD at pH 9 and the corresponding simulation with two components at 50:50%. **h** Individual components of the simulated spectrum: component 1 corresponds to a radical localized in the CD domains, while component 2 is attributed to the radical confined in the nitrogen-rich porphyrinoid sites (details are given in supplementary information).

pH-dependence agrees well with the structural characterization and underscores the role of protonation–deprotonation equilibria in modulating the $Cu^{2+}$ axial coordination and enabling a site-specific electron transfer. Overall, the photo-induced electron cascade in CuP@CD mimics the enzymatic electron transfer in CcO. In this cascade, CD acts as the light harvester and triggers the electron transfer to a CuP site, where a two-electron reduction of oxygen occurs.

**Light-driven ORR: Unraveling the oxygen reduction reaction pathway in CuP@CD**
Early on, the ORR activity of CuP@CD was evaluated by cyclic voltammetry (CV) investigations. The highest activity in terms of electro-catalytic currents at around −0.5 V vs. Ag/AgCl was noted at a basic pH (pH 9) during the cathodic scans (Fig. S16). The overall ORR activity under air was amplified when pure $O_2$ was utilized (Fig. S17). Light-driven ORR was independently explored by means of LEPR spectroscopy. With in-situ measurements, we tracked the evolution of photo-excited species in a mixture of CuP@CD, TEOA, and N-tert-butyl-α-phenylnitrone (PBN) as a spin trap for hydroperoxyl radicals (˙OOH) at pH 9. 2D color maps (Fig. 4a) recorded under light-off and light-on sequences revealed the formation of ˙OOH radicals immediately following photo-irradiation (Fig. 4b). Simulated hyperfine coupling constants ($A_N = 1.45$ mT, $A_H = 0.28$ mT) identified ˙OOH radicals, which is consistent with an $H_2O_2$ production (Fig. S18)[27,50]. A lack of ˙OOH radicals when probing $H_2$P@CD highlights the essential role of $Cu^{2+}$ coordination in ORR activity (Fig. S19) and aligns with the CV results.

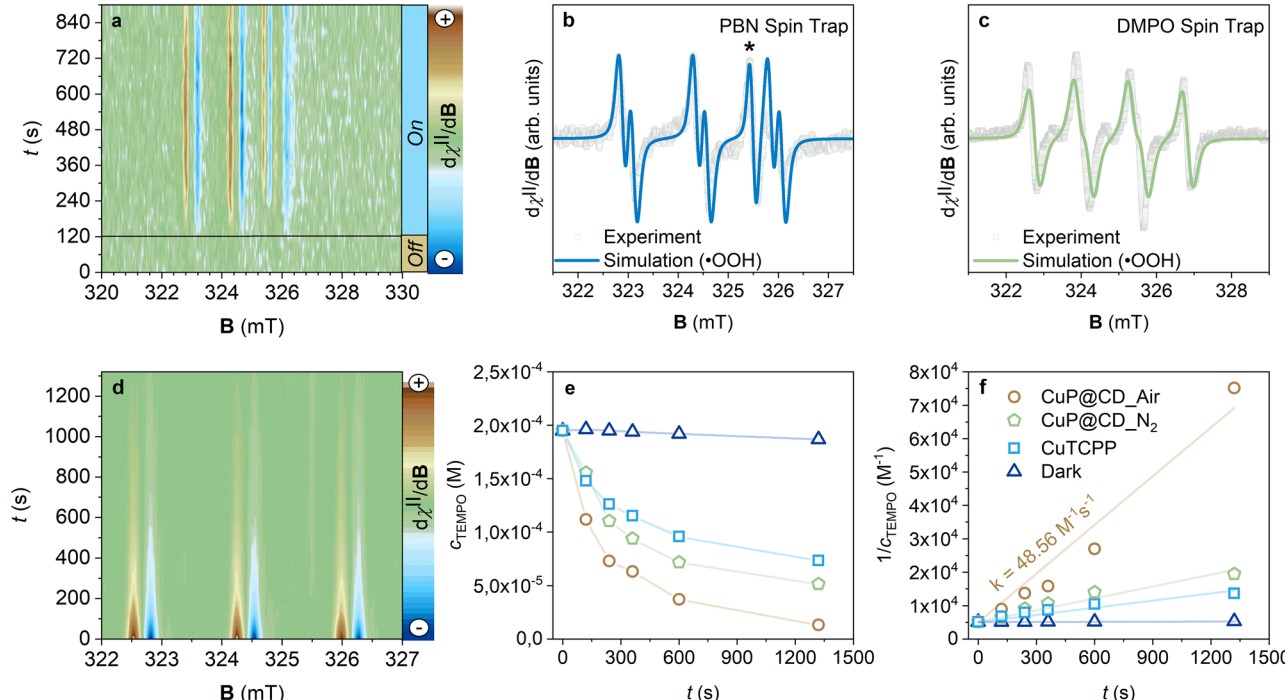

**Fig. 4 | Photoinduced radical intermediates reveal the oxygen-reduction reaction pathway in CuP@CD. a** 2D color map obtained from in-situ light-induced electron paramagnetic resonance (LEPR) experiments (continuous-wave (CW) X-band, 9.07 – 9.08 GHz, $T = 293$ K, @325 nm, 40 mW), illustrating the time evolution of the EPR signals during a dynamic light excitation sequence (light-off and light-on) of CuP@CD in water (pH 9) containing triethanolamine (TEOA, 0.1 M) and α-phenyl-N-tert-butylnitrone (PBN) as a spin trap. **b** Experimental data (grey symbols) and computer simulation (blue line) of PBN spin-trapping LEPR spectra of CuP@CD revealing the presence of hydroperoxyl radicals (·OOH). The star symbol represents the signal of photo-induced electrons on CuP@CD. **c** Experimental data (grey

symbols) and computer simulation (green line) of 5,5-dimethyl-1-pyrroline-N-oxide (DMPO) spin-trapping LEPR spectra of CuP@CD, confirming the presence of (·OOH) radicals. **d** 2D color map from in-situ LEPR experiments (CW X-band, 9.07 – 9.08 GHz, $T = 293$ K), showing the time evolution of LEPR signals during photo-excitation of CuP@CD in water (pH 9) containing TEOA (0.1 M) and 2,2,6,6-tetra-methylpiperidine-1-oxyl (TEMPO, $1.95 \times 10^{-4}$ M). **e** Decay of the TEMPO radical LEPR signal ($T = 293$ K) as a function of photo-irradiation time (@325 nm) under conditions presented in the legend of (**f**). **f** Second-order kinetic analysis of the TEMPO signal decay using the dataset presented in (**e**).

To pinpoint the oxygen binding site, EPR spectra of CuP@CD were taken under $N_2$- and $O_2$-saturated conditions. Important is that the $Cu^{2+}$ fingerprints fade away in an $O_2$-rich environment. Here, it is an accelerated spin relaxation upon oxygen binding that broadens the resonance signal. We consider this as support for the localization of the reaction at the CuP sites (Fig. S20). Control experiments confirmed the light-induced nature of $Cu^{2+}$ reduction. For example, EPR spectra recorded in the dark before and after TEOA addition displayed the characteristics of $Cu^{2+}$ (Fig. S21).

Next, LEPR measurements with 5,5-dimethyl-1-pyrroline N-oxide (DMPO), a spin trap to distinguish between ·OOH and ·OH radicals, confirmed the absence of ·OH by means of hyperfine coupling constants ($A_N = 1.20$ mT, $A_H = 1.18$ mT)[50]. Excluding a four-electron reduction pathway or a Fenton-like/electron-driven $H_2O_2$ decomposition, confirms $H_2O_2$ as the final product of ORR for CuP@CD (Fig. 4c). Notable, TEOA mimicking the role of cytochrome $c$ suppresses any hole-induced decomposition and ensures $H_2O_2$ stability even in the absence of ·OH radicals[51]. Considering that ·OOH radicals were trapped and distinct electro-catalytic currents were present at around −0.5 V vs. Ag/AgCl, $H_2O_2$ is produced via an indirect two-step, single-electron reduction pathway (Eq. 1 and 4 in the SI)[46].

$H_2O_2$ quantification was attempted using commercial detection kits (Spectroquant $H_2O_2$ Test and Peroxide Assay Kit-Sigma Aldrich). False-positive signals due to TEOA interference precluded, however, any reliable interpretation. Potassium iodide-based spectroscopic detection confirmed, on one hand, $H_2O_2$ production under solar irradiation but, on the other hand, lacked specificity due to interference with TEOA (Fig. S22). Titrimetric methods proved also to be

inconclusive due to a green coloration, which interfered with an accurate endpoint determination[51]. To overcome any of the afore-mentioned limitations, LEPR measurements employing 2,2,6,6-tetra-methylpiperidine-N-oxyl radical (TEMPO) were conducted. TEMPO is oxidized by ·OOH as well as ·OH radicals to yield a diamagnetic and EPR-silent oxoammonium cation[52,53]. Both PBN and DMPO spin trap experiments – *vide supra* – excluded the presence of ·OH radicals. Therefore, the decrease in the TEMPO EPR signal intensity is directly proportional to the photo-production of $H_2O_2$[8]. Importantly, CuP@CD exhibited no detectable activity in the dark (Fig. S23). Upon photo-irradiation, a rapid TEMPO decrease corroborated the formation of ·OOH radicals en-route towards $H_2O_2$ production (Fig. 4d, Fig. S24). Kinetic analysis of the LEPR spectra (Fig. 4e, f) by second-order kinetic model revealed that CuTCPP exhibited significantly lower ORR activity than CuP@CD. This comparison is vital as it emphasizes the crucial role of the CD domain's reductive force in replicating $CcO$ functionality.

Mitigated $H_2O_2$ production under $N_2$-saturated conditions supports the indirect two-step, single-electron reduction ORR pathway. In ambient air, $H_2O_2$ production rate is 7.9 mmol·g$^{-1}$·mL$^{-1}$·h$^{-1}$, as determined from kinetic analyses – details are gathered in the Supporting Information. At this stage, we cannot rule out a parallel radical recombination pathway between two ·OOH radicals confined at the active sites (Eq. 3 in the SI). Considering $2\cdot OOH \rightarrow H_2O_2 + O_2$, the effective $H_2O_2$ production rate would be 3.95 mmol g$^{-1}$ mL$^{-1}$ h$^{-1}$. Regardless of 7.9 or 3.95 mmol g$^{-1}$ mL$^{-1}$ h$^{-1}$ for the earlier or later path-ways, respectively, these $H_2O_2$ photo-production rates are competitive with those reported for state-of-the-art photocatalysts (Table S3).

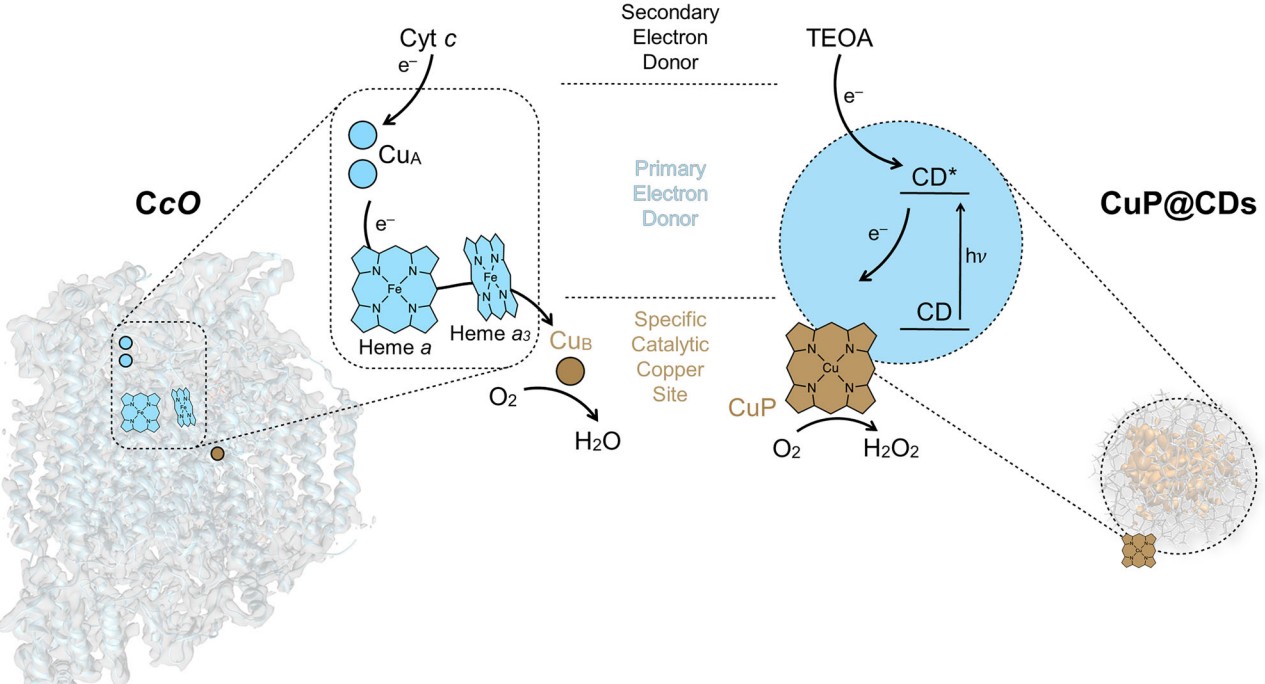

**Fig. 5 | Single-atom-engineered carbon dots mimic the site-specific electron-transfer cascade of cytochrome c oxidase (CcO).** Photoexcitation of carbon dots (CDs) triggers electron transfer to the catalytic Cu-porphyrin (CuP) site, mimicking the sequence from CuA via heme $a/a_3$ to the catalytic CuB site in CcO. In CuP, as in CuB, molecular oxygen is reduced to form $H_2O_2$. The secondary electron donor triethanolamine (TEOA) functions analogously to cytochrome c (Cyt c) in CcO, completing the photocatalytic electron circuit in CuP@CDs. Cryo-EM density map of human cytochrome CcO (CIV) was taken from ref. 66, and the representative carbon-dot model from ref. 67.

Notably, the fact that 71.2% of dissolved $O_2$ was converted to $H_2O_2$ within one hour–based on a two-step, single-electron reduction ORR pathway–confirms an effective consumption of $O_2$. The apparent quantum yield (AQY), calculated based on a photon-to-electron conversion at 325 nm excitation, was 1.5%.

Sole TEOA had only a subtle impact on the reaction kinetics (Fig. S28). But its absence resulted in charge accumulation and eventually in CuP@CD degradation[54]. It highlights its role as an electron donor analogous to cytochrome c as part of the CcO mimicry (Fig. 5, Fig. S29). Stability tests with CuP@CD under photocatalytic conditions were ran by means of XPS and EPR measurements (Fig. S30, S31) and showed no significant degradation.

Overall, these results highlight the functional synergy between CDs as electron reservoirs and $Cu^{2+}$ as catalytic sites to set up an enzyme-like spatial and redox organization. To this end, an electron transfer upon photoexcitation occurs exclusively at the CuP sites, which leaves the function of the CDs to act as a light-harvester and electron-donor. By combining spatial compartmentalization with functional differentiation, a highly directional electron transfer cascade is established. This is very similar to what is known for CcO, where individual cofactors facilitate a series of redox reactions. A biomimetic arrangement like that in CuP@CD accelerates charge separation, suppresses charge recombination, and fosters high $H_2O_2$ production rates. It thereby distinguishes it from more traditional photocatalytic systems that lack structural order.

## Discussion

A highly efficient enzymatic-like photocatalyst, inspired by cytochrome c oxidase, has been developed for light-driven ORR by utilizing non-toxic and earth-abundant single-atom-engineered CDs. It mimics CcO's site-specific electron transfer through its dual active sites, enabling the selective reduction of oxygen to $H_2O_2$ under ambient conditions. This enzymatic-inspired design establishes a new paradigm for precise enzymatic photocatalysis. It closes the gap between natural enzymatic processes and synthetic photocatalysis for transformative applications in energy and environmental technologies.

## Methods

### Chemicals and reagents

All chemicals were supplied by Sigma-Aldrich and used without further purification unless otherwise stated. The materials employed for the synthesis had the following purities: formamide (ReagentPlus®, ≥99.0%, GC), L-glutathione (reduced, ≥98.0%), and copper(II) nitrate hydrate (99.999%, trace metals basis). Dialysis tubing (2 kDa cut-off, benzoylated) for colloid purification was purchased from Sigma-Aldrich. Biotech CE tubing (0.5 kDa cut-off) was purchased from Spectrum Laboratories, Inc.

### Synthesis of $H_2P@CD$

$H_2P@CD$ were prepared following the method described in the literature[55]. Briefly, 307 mg of L-glutathione were dissolved in 10 mL of formamide under sonication until a clear solution was obtained. The mixture was then subjected to microwave heating at maximum power for 40 seconds. $H_2P@CD$ were passed through a syringe filter and then dialyzed (2 kDa cut-off) against deionized water for 72 h to remove residual small molecular species.

### Synthesis of CuP@CD

For this purpose, an aqueous suspension of 10 mL containing 15 mg of $H_2P@CD$ was mixed with 1 mL of a 0.04 M Copper(II) nitrate hydrate solution. The mixture was stirred vigorously for 24 h at room temperature in the dark. This corresponds to a theoretical copper loading of ~15 wt%. The exact Cu content after dialysis was determined by XPS analysis. The dispersion of $H_2P@CD$ with immobilized single atoms was then purified by dialysis using a cellulose membrane with a 0.5 kDa pore size to remove unbound $Cu^{2+}$ ions. Size-exclusion column

chromatography (PD MiniTrap G-25) was performed to confirm the monodispersity of CuP@CD.

## Synthesis of CuTCPP

TCPP sample was available from previous study[56]. Briefly, 1.5 mg of TCPP was dissolved in 1 mL of water and sonicated for 15 min. Subsequently, 0.5 mL of a 0.008 M Copper(II) nitrate hydrate solution was added to the mixture. The resulting solution was stirred vigorously at room temperature in the dark for 24 h.

## Buffer preparation

Buffers for pH-dependent measurements were prepared using 0.1 M aqueous solutions. A carbonate-bicarbonate buffer was prepared by mixing sodium bicarbonate and anhydrous sodium carbonate (pH 9). A potassium phosphate buffer was prepared using monobasic dihydrogen phosphate and dibasic monohydrate phosphate (pH 7). A citrate buffer was prepared by combining sodium citrate dihydrate and citric acid monohydrate (pH 4).

## Instrumentation

The materials, $H_2P@CD$ and CuP@CD, were characterized using a JEM 2010 TEM instrument (Jeol, Japan). A diluted aqueous dispersion of the material was prepared and mixed with isopropanol to enhance wettability. To reduce aggregation caused by the presence of alcohol, the dispersion was filtered through a 0.45 μm filter. The final mixture was deposited onto a carbon-coated gold grid and dried at room temperature for 24 h.

Absorption spectra were collected using a Shimadzu UV-1900i spectrometer. Steady-state emission spectroscopy was performed with an Edinburgh FS5 spectrofluorimeter. Emission lifetimes and time-resolved emission spectra (TRES) were measured by time-correlated single-photon counting. We employed either a Fluorolog 3 system (Horiba Jobin Yvon) driven by a SuperK Fianium FIU6PP supercontinuum laser (NKT Photonics) with an R3809U-50 microchannel-plate photomultiplier (Hamamatsu), or an Edinburgh FS5 spectrofluorimeter equipped with a VisUV picosecond laser head (PicoQuant, pulse width <85 ps).

Ultrafast transient absorption experiments were conducted using an Astrella-F-1K amplified Ti:sapphire femtosecond laser system (Coherent) operating at a repetition rate of 1 kHz, with a pulse energy of 5 mJ and a pulse duration of 80 fs. The Helios TA pump/probe detection system (Ultrafast Systems) was used for data acquisition. The white-light probe continuum was produced by focusing a small portion of the 800 nm fundamental beam into a 2 mm $CaF_2$ plate mounted on a translating stage. Pump pulses at the selected excitation wavelength were generated by sending ~1.2 mJ of the amplifier output into a TOPAS Prime optical parametric amplifier (Light Conversion) equipped with a NirUVis extension. A depolarizing optic in the pump beam minimized rotational anisotropy effects, and interference filters with 5–10 nm bandwidth were used to define the excitation wavelength and suppress residual 800 nm light. Samples were contained in sealed 2 mm quartz cuvettes under argon, with optical densities at the pump wavelength adjusted to 0.5–0.7[57]. Transient absorption data were analyzed following established protocols[58]. Initial inspection used singular value decomposition (SVD) and global fitting with a sequential kinetic model to obtain evolution-associated spectra (EAS) that describe the time-dependent decay of the excited-state manifold. To extract spectra that can be assigned to distinct photophysical species, we then applied target analysis with physically motivated kinetic schemes, yielding species-associated spectra (SAS). Global and target fits, as well as SVD, were carried out using the R-based packages TIMP and GloTarAn[59,60].

FT-IR spectra were recorded on an iS50 FT-IR spectrometer (Thermo Nicolet) using diamond ATR accessory. Briefly, a droplet of a water dispersion of the relevant material was placed on the diamond crystal and dried.

XPS measurements were performed using a Nexsa G2 spectrometer (Thermo Fisher Scientific) equipped with an Al Kα source (12 kV, 100 μm spot size). Survey spectra were acquired with a pass energy of 150 eV and a step size of 1 eV, while high-resolution (HR-XPS) spectra were obtained with a pass energy of 50 eV and a step size of 0.1 eV. Elemental analysis was conducted using high-resolution scans of the respective elements. Data evaluation and peak deconvolution were carried out using the Avantage software package (Thermo Fisher Scientific). The spectral analysis included Shirley background subtraction and peak fitting using mixed Gaussian–Lorentzian functions. All binding energies were referenced to the C–C bond at 284.8 eV.

High-resolution transmission electron microscopy (HR-TEM) images were acquired using a TITAN 60-300 HR-TEM microscope equipped with an X-FEG emission gun, operating at an acceleration voltage of 300 kV. Scanning transmission electron microscopy high-angle annular dark-field (STEM-HAADF) imaging for energy-dispersive X-ray spectroscopy (EDS) elemental mapping was performed using a FEI Titan HR-TEM microscope operating at 80 kV. Sample preparation followed the same procedure as for TEM measurements: a diluted aqueous dispersion of the material was mixed with isopropanol to enhance wettability, filtered through a 0.45 μm filter to minimize aggregation, and deposited onto a carbon-coated copper grid. The sample was then dried at room temperature for 24 h.

All electrochemical measurements were performed using a Metrohm μAutolab FraIII potentiostat. For pH-dependent cyclic voltammetry (CV) measurements, a glassy working electrode was employed, with a platinum wire serving as the counter electrode and an Ag/AgCl (3.5 M NaCl) reference electrode.

Electron paramagnetic resonance (EPR) spectra were recorded on a JEOL JES-X-320 spectrometer operating at X-band ($\approx 9.07-9.08$ GHz) equipped with an ES-CT470 variable-temperature unit (He/$N_2$ cryostat). Unless noted otherwise, measurements were carried out at 80 K with cavity Q-factors above 6000. Samples were placed in Suprasil quartz tubes (Wilmad, optical density ≤ 0.5). The magnetic field scale was calibrated using a standard with $g_{eff} = 2.00105$[61]. To avoid power saturation effects, the microwave power was set to 1.9 mW, with a modulation width of 0.35 mT and a modulation frequency of 100 kHz. All spectra were collected with a time constant of 30 ms and a sweep time of 2 min, using three accumulations to enhance the signal-to-noise ratio. A HeCd laser (325 nm, 200 mW) from Kimmon Koha Co. Ltd. served as the UV light source, equipped with a fiber optic cable coupling the light directly into the cavity resonator via its dedicated window. EPR spectra were simulated using the EasySpin package on the Matlab software platform for spin-Hamiltonian modeling[62].

X-ray Absorption Spectroscopy. Data were recorded at SAMBA beamline in fluorescence mode (CuP@CD) with HPGe detector (Mirion) and Si (220) monochromator. Standard CuTCPP was measured in transmission mode. Beamline was calibrated on copper foil as per standard procedures at the maximum of the first derivative (8979 eV). $S_0^2$ value has been determined (0.92) by fitting the first coordination shell of a standard Cu porphyrin sample and it is in line with recent literature[63]. The theoretical standards have been obtained from $F_{eff}$ 8.4[64] and computed on a copper porphyrin structure from[65]. Fits have been performed from 3 to 11 Å$^{-1}$ in r-space on a $k^3$ weighted EXAFS signal.

## Photocatalytic $H_2O_2$ production

Photocatalytic $H_2O_2$ production experiments were carried out using a xenon lamp equipped with an AM1.5 G filter, simulating one sun intensity (100 mW cm$^{-2}$) as the irradiation source. A colloidal suspension of CuP@CD (0.5 mg) was dispersed in a total volume of 6 mL of water containing TEOA (pH9) as the sacrificial electron donor.

Light-driven ORR for $H_2O_2$ generation was monitored via LEPR spectroscopy (CW X-band, 9.07–9.08 GHz, $T = 293$ K) under photo-excitation at 325 nm (40 mW cm$^{-2}$). The optical power density was measured using an external digital power meter (Thorlabs PM100D) positioned at the sample plane. For these measurements, CuP@CD (0.024 mg) was suspended in 120 μL of aqueous TEOA solution (0.1 M, pH9) containing TEMPO ($1.95 \times 10^{-4}$ M). For further details see Supporting Information. Due to the complexity of the in-situ detection setup, full statistical replication was not performed. The reported production rate was obtained from a single kinetic measurement; therefore, no standard deviation is provided. The quantification protocol is detailed in the Supporting Information (page 28–31). No other reaction products were detected within the sensitivity limits of EPR analysis (Fig. 4b, c).

## Data availability
The data supporting the findings of this study are available in the Zenodo repository under DOI: 10.5281/zenodo.17398465 and are also provided in Source Data and Supplementary Data. Source data are provided with this paper.

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

## Acknowledgements

This article has been produced with the financial support of the European Union under the REFRESH – Research Excellence For Region Sustainability and High-tech Industries project number CZ.10.03.01/00/22_003/0000048 via the Operational Programme Just Transition – (R.Z). The work was also supported by the ERDF/ESF project TECHSCALE (No. CZ.02.01.01/00/22_008/0004587) – (D.P.). A.C. is a member of the research staff of CONICET. D.M.G. acknowledges the support from "Solar Technologies go Hybrid" an initiative from the free state of Bavaria, and from the DFG 517/29-1. G.Z. acknowledges the support from Horizon-Europe EIC Pathfinder Open 2023 - GlaS-A-Fuels (project No 101130717). Š.K. gratefully acknowledge support by the Czech Science Foundation, project GA CR 23-07971S. The work was further supported by the European Union's Horizon 2020 project SAN4Fuel (No. 101079384, Horizon-Widera-2021) – (Š.K). Authors also acknowledge the support from the Horizon Europe project HORIZON-WIDERA-2022-TALENTS "APPROACH" (No 101120397) – (L.Z., Z.B.). Financial support from the Hertha und Helmut Schmauser-Stiftung is acknowledged (A.C.). The authors gratefully thank J. Stráská and O. Tomanec for their expert assistance with electron microscopy measurements. E.F. acknowledges Synchrotron SOLEIL for provision of beamtime at SAMBA.

## Author contributions

L.Z. was involved with the investigation, analysis, writing of the original draft, methodology and visualization. G.Z. was involved with the investigation, analysis, editing during writing, methodology and visualization. A.G.G. and H.K. were involved with the analysis. C.O. and M.A.H. were involved in the synthesis and analysis. Z.B., D.P. and E.F. were involved with the investigation and analysis. Š.K. was involved with the review and editing during writing. A.C. was involved with the supervision, investigation, analysis, review and editing during writing. R.Z. and D.M.G. were involved with the supervision, funding acquisition and review and editing during writing.

## Funding

## Competing interests

The authors declare no competing interests.
