## [Transparent Peer Review file · Nature Communications]

Enzyme-Inspired Single-Atom Photocatalysis for Oxygen Reduction to Hydrogen Peroxide

Corresponding Author: Professor Dirk Guldi

Version 0:

Reviewer comments:

Reviewer #1

(Remarks to the Author)

This paper presents the design of single-atom-engineered carbon dots (CDs) that effectively mimic the function of cytochrome c oxidase (CcO). The study offers valuable insights into the electron transfer processes following photoexcitation of the CDs, particularly from the CDs to the copper-containing active site (CuP) and the sacrificial electron donor, triethanolamine (TEOA). Owing to its depth, novelty, and relevance, this work holds significant potential to advance interdisciplinary research in the field of advanced photocatalysis, but major revisions are recommended prior to acceptance.

1. In this work, CDs were synthesized via hydrothermal methods using glutathione and formamide and it is expected that the different types of side products such as organic molecules and different emitting CDs can be formed during the time of synthesis. In this work, the crude product is purified via syringe filtration and then dialysis which is not sufficient for the purification of small molecules and different emitting CDs. Thus, it needs proper separation and purification of the crude product either column chromatography or HPLC.

2. The origin of fluorescence in CDs is mentioned here coming out from two different regions: one is CDs domains, and another is from porphyrinoid sites. But CDs have complex and heterogeneous structures. The origin of PL of C-Dots is due to two

different emissive species within C-Dots: one is the core state, and another is surface state. Thus need a justification of proof about the present of porphyrinoid site after the proper purification of C-Dots. In this stage, the conclusion of the work is not strong enough, it's basically an assumption.

3. The two PL maxima of HP@CD at 680 nm and 650 nm are not clearly visible in Figure 1b. Moreover, Figure 1b and 1c suggest that the HP@CD exhibits a PL maximum at 680 nm, which shifts to 650 nm upon the formation of CuP@CD. Therefore, a proper PL spectrum with an appropriate excitation wavelength is required to better investigate this behavior.

4. Figure S3 (a–b) shows emission color maps of H2P@CD, clearly indicating quenching of PL intensity upon the addition of Cu²⁺ ions in aqueous solution. However, the main manuscript claims that 'the fluorescence centered on CD in CuP@CD remains mostly unaffected.' A proper explanation is needed to justify this apparent discrepancy.

5. Transient absorption spectroscopy (fsTAS) measurements were performed using 420 nm excitation, which primarily probes the surface states. However, to obtain a clearer understanding of the carrier dynamics within the CD domain, fsTAS should be conducted using 340 nm excitation, which corresponds to the excitation of the CD core. Some important papers are suggested in ultrafast dynamics of CDs, such as *Nanoscale*, 2022, 14, 15812-15820, *Nanoscale*, 2024, 16, 8143-8150.

6. The mechanism of photo-induced electron transfer from CuP@CD to various sacrificial electron donors is not clearly explained. A diagram illustrating the energy levels of the CDs along with those of the quenchers is needed to better elucidate the electron transfer mechanism.

7. There are several typographical errors throughout the main text and SI. The authors are advised to thoroughly check the entire document.

Reviewer #2

(Remarks to the Author)

The author successfully developed the first copper monomer-enhanced carbon dot (CDs) enzyme-like photocatalyst. This material precisely mimics the electron transfer cascade of cytochrome c oxidase (CcO), achieving high selectivity in the photocatalytic reduction of oxygen to generate H₂O₂ at room temperature. Its activity surpasses that of all known metal-based catalysts. There are some interesting results; however, there still exist some problems that need to be corrected and

explained. The following are some points that might be useful to refine the article.

1. The current title fails to fully reflect the application breakthrough of the work. It is suggested that the relevant applications be highlighted in the title.
2. The three distinct fluorescent CD areas respectively refer to which three? It is recommended to provide more detailed explanations in the text. No obvious single Cu atom bright spots were observed in the HAADF-STEM images.
3. The EDS only shows the Cu mapping, lacks the comparison of C/N/O carrier elements, and the resolution is insufficient to conclude that the Cu monatomic species are uniformly distributed on the carbon dots.
4. The author merely emphasized the absence of the Cu⁺/Cu⁰ characteristic peaks, but did not provide clear evidence for the existence of Cu²⁺.
5. What is the reason for the decrease in the Cu 2p XPS binding energy of CuP@CD compared to CuTCCP? Is it due to axial coordination? It is recommended to provide a detailed explanation of the cause of this change.
6. How to calculate the yield of hydrogen peroxide generated in the CuP@CD photocatalytic ORR process? The authors did not clearly explain the experimental basis for calculating the H₂O₂ yield. They only relied on the TEMPO-EPR spectrum for semi-quantitative analysis. However, this method lacks a standard concentration-signal intensity calibration curve, and the linear relationship between the TEMPO⁺ signal and the H₂O₂ concentration has not been verified.
7. What is the main intrinsic reason why CuP@CD outperforms other photocatalysts? How does the specific site electron transfer of CcO promote efficient conversion?
8. What is the conversion rate of H₂O₂ produced by CuP@CD in photocatalysis, what is the apparent quantum yield (AQY), and how is the cycle stability?
9. The experimental details of the light-driven ORR for the production of H₂O₂ need to be provided in greater detail.
10. How does the CuP@CD photocatalyst perform in the actual production of H₂O₂ through photocatalysis? Why not adopt other methods to quantify H₂O₂?

Version 1:

Reviewer comments:

Reviewer #1

(Remarks to the Author)

The authors have satisfactorily addressed nearly all of my concerns as a reviewer, and the revised manuscript is suitable for publication.

Reviewer #2

(Remarks to the Author)

The authors have revised the manuscript accordingly. I would recommend it for publication.

Open Access This Peer Review File is licensed under a Creative Commons Attribution 4.0 International License, which permits use, sharing, adaptation, distribution and reproduction in any medium or format, as long as you give appropriate credit to the original author(s) and the source, provide a link to the Creative Commons license, and indicate if changes were

made.

Reviewer #1

Reviewer's Comment:

This paper presents the design of single-atom-engineered carbon dots (CDs) that effectively mimic the function of cytochrome c oxidase (CcO). The study offers valuable insights into the electron transfer processes following photoexcitation of the CDs, particularly from the CDs to the copper-containing active site (CuP) and the sacrificial electron donor, triethanolamine (TEOA). Owing to its depth, novelty, and relevance, this work holds significant potential to advance interdisciplinary research in the field of advanced photocatalysis, but major revisions are recommended prior to acceptance.

Reply to the Reviewer's Comment:

We highly appreciate the positive evaluation of our work and insightful comments, which have helped us a lot to improve our manuscript.

Reviewer's Point #1:

In this work, CDs were synthesized via hydrothermal methods using glutathione and formamide and it is expected that the different types of side products such as organic molecules and different emitting CDs can be formed during the time of synthesis. In this work, the crude product is purified via syringe filtration and then dialysis which is not sufficient for the purification of small molecules and different emitting CDs. Thus, it needs proper separation and purification of the crude product either column chromatography or HPLC.

Reply to the Reviewer's Point #1:

We thank the reviewer for raising the important point regarding the complexity of CDs synthesis and the potential presence of fluorescent side products. To directly address this concern, we have performed size-exclusion column chromatography of CuP@CD. The sample eluted as a single-colored broad band, consistent with a dominant high-molecular-weight fraction. Four sequential aliquots (labeled F1–F4) have been collected during the elution, continuing until the colored fraction was no longer visible. Each of them, that is, F1–F4 has been analyzed by PL spectroscopy following 355 nm photoexcitation (Figure S2c). All sub-fractions exhibited dual-emission features, corresponding to the CD domains at 450 nm and the porphyrinoid sites at 650 nm. What has been of interest is that the relative intensity ratio (I_{650}/I_{450}) has remained statistically invariant across the fractions (Figure S2d). This strongly supports our notion that the porphyrinoid emission originates from strongly integrated structures, rather than from separable molecular side products.

Changes in the revised versions of the manuscript and supporting information:

(pg. 6) "Uniformity of CuP@CD was independently confirmed by size-exclusion column chromatography. Essentially, identical emissions were detected for all fractions (Figure S2c, d). This rules out that porphyrinoid species are not integrated parts of either H₂P@CD or CuP@CD."

(pg. 19) "Size-exclusion column chromatography (PD MiniTrap G-25) was performed to confirm the monodispersity of CuP@CD."

Figure S2. (a) Emission spectra of H₂P@CD under different photo-excitation wavelengths. (b) Emission spectra ($\lambda_{\text{ex}} = 420 \text{ nm}$) comparing H₂P@CD and CuP@CD. (c) Emission spectra of F1-F4 fractions of CuP@CD collected after column chromatography ($\lambda_{\text{ex}} = 355 \text{ nm}$). (d) Ratio of integrated PL intensities at 650 nm and 450 nm across fractions.

Reviewer's Point #2:

The origin of fluorescence in CDs is mentioned here coming out from two different regions: one is CDs domains, and another is from porphyrinoid sites. But CDs have complex and heterogeneous structures. The origin of PL of C-Dots is due to two different emissive species within C-Dots: one is the core state, and another is surface state. Thus need a justification of proof about the present of porphyrinoid site after the proper purification of C-Dots. In this stage, the conclusion of the work is not strong enough, it's basically an assumption.

Reply to the Reviewer's Point #2:

We thank the reviewer for raising this critical point of our investigation. We agree with the reviewer that the PL of CDs might have different origin, including core and surface states. In our previous work (e.g., ACS Nano 2021, 10.1021/acsnano.0c09781; JACS 2022, 10.1021/jacs.1c07049; Adv. Optical Mater. 2023, 10.1002/adom.202300750; Small 2023, 10.1002/smll.202207238 and Angew. Chem. 2025, 10.1002/anie.202418626) we have, however, consistently demonstrated that the dominating features are due to entrapped molecular chromophores/fluorophores rather than from the graphitic core states or surface defects. That said, we fully recognize that both interpretations are applicable for CDs but it depends on the CD synthesis route and the corresponding precursors. To provide independent evidence for this, we have performed a diagnostic quenching experiment based on our previous study (ACS Nano 2021, 10.1021/acsnano.0c09781). In this experiment, the emission assigned to the molecular fluorophores was substantially quenched upon freezing, whereas the CD core emission remained unaffected. This matches the characteristics of embedded molecular fluorophores and further supports our structural assignment (Figure R1).

Additional support for the presence of porphyrinoids integrated into CDs: First, the absorption spectrum of H₂P@CD exhibits one Soret-band and four Q-bands. Upon coordination with Cu²⁺, these Q-bands merge into two, reflecting the expected symmetry transformation upon metalation (Figure 1a). Second, EPR spectroscopy of CuP@CD confirms a Cu²⁺ species coordinated in-plane to four nitrogen atoms, consistent with a porphyrin-like ligand field (Figure 3g, h; simulation details are provided in SI, page 3). Third, extended X-ray absorption fine structure (EXAFS) measurements revealed bond distances that are nearly identical to those seen in a CuTCPP reference (Table S1).

Figure R1. Influence of liquid-to-solid (water-to-ice) phase transition in the solvent (water) on fluorescence of H₂P@CD.

Reviewer's Point #3:

The two PL maxima of HP@CD at 680 nm and 650 nm are not clearly visible in Figure 1b. Moreover, Figure 1b and 1c suggest that the HP@CD exhibits a PL maximum at 680 nm, which shifts to 650 nm upon the formation of CuP@CD. Therefore, a proper PL spectrum with an appropriate excitation wavelength is required to better investigate this behavior.

Reply to the Reviewer's Point #3:

We thank the reviewer for pointing this out. In the revised version of the manuscript, we have clarified the distinction between the two porphyrinoid emissions and have presented a more detailed PL spectra under optimized excitation ($\lambda_{\text{ex}} = 420 \text{ nm}$) in Figure S2b. The new spectra clearly resolve both emissions, that is, at 680 and 650 nm for H₂P@CD and CuP@CD, respectively, and support our interpretation of selective Cu²⁺ coordination to P680.

Changes in the revised versions of the manuscript and supporting information:

(pg. 5) "Among others, Cu²⁺ coordination causes a quenching of the 680 nm fluorescence and lifetime thereof, but not of the 650 nm fluorescence (Figure 1e, g, and S2b)."

Figure S2. (a) Emission spectra of H₂P@CD under different photo-excitation wavelengths. (b) Emission spectra ($\lambda_{\text{ex}} = 420 \text{ nm}$) comparing H₂P@CD and CuP@CD. (c) Emission spectra of F1-F4 fractions of CuP@CD collected after column chromatography ($\lambda_{\text{ex}} = 355 \text{ nm}$). (d) Ratio of integrated PL intensities at 650 nm and 450 nm across fractions.

Reviewer's Point #4:

Figure S3 (a–b) shows emission color maps of H₂P@CD, clearly indicating quenching of PL intensity upon the addition of Cu²⁺ ions in aqueous solution. However, the main manuscript claims that 'the fluorescence centered on CD in CuP@CD remains mostly unaffected.' A proper explanation is needed to justify this apparent discrepancy.

Reply to the Reviewer's Point #4:

We thank the reviewer for this important observation. As correctly noted, both emission domains show different degrees of quenching upon Cu^{2+} addition. New concentration-dependent, integrated PL analyses (Figure S3e) have clarified that the porphyrinoid-centered emission at 680 nm is subject to a stronger quenching than the CD-centered fluorescence at 450 nm. This is particularly evident at low Cu^{2+} concentrations. All together this supports the interpretation that Cu^{2+} coordination preferentially affects the porphyrinoids. We have modified the revised version of the manuscript accordingly and have included the integrated PL data in the revised version of the supporting information.

Changes in the revised versions of the manuscript and supporting information:

(pg. 5) "A closer look at the fluorescence reveals that Cu^{2+} coordination quenches the one of the porphyrinoid domains, while the impact on the CD-centered emission is subtle (Figure 1c)."

Figure S3. (a–b) Emission color maps of $\text{H}_2\text{P@CD}$ in aqueous solutions with varying Cu^{2+} concentrations. (c–d) Normalized time-resolved fluorescence intensity ($\lambda_{\text{ex}} = 372$ nm) at the emission maxima of carbon domains ($\lambda_{\text{em}} = 450$ nm) and porphyrinoid centers ($\lambda_{\text{em}} = 650$ nm) as a function of Cu^{2+} concentration (pH 9). (e) Integrated fluorescence intensity as a function of Cu^{2+} concentration for both CD and CuP sites, respectively.

Reviewer’s Point #5:

Transient absorption spectroscopy (fsTAS) measurements were performed using 420 nm excitation, which primarily probes the surface states. However, to obtain a clearer understanding of the carrier dynamics within the CD domain, fsTAS should be conducted using 340 nm excitation, which corresponds to the excitation of the CD core. Some important papers are suggested in ultrafast dynamics of CDs, such as *Nanoscale*, 2022, 14, 15812-15820, *Nanoscale*, 2024, 16, 8143-8150.

Reply to the Reviewer’s Point #5:

We thank the reviewer for this suggestion. We have performed fsTAS experiments using 350 nm in the absence and in the presence of copper ions, with the aim of obtaining indications of the carrier dynamics upon CD core excitation. Unfortunately, although we targeted CD excitation, CD-centered transient-absorptions signals were too weak, as commonly observed for other types of CDs. Instead, we obtained transient spectra largely dominated by the porphyrinic sites, as indicated by the negative signals at around 400 and 650 nm and the positive signals between 450 and 600 nm. We have included these experiments in the revised version of the supporting information and have added the suggested references to the revised version of the manuscript.

Changes in the revised versions of the manuscript and supporting information:

(pg. 6) “Photoexcitation of $\text{H}_2\text{P@CD}$ and CuP@CD at, for example, 350 nm was also explored to investigate the excited-state dynamics upon CD-based excitation.^{33,34} The transient spectra are, however, even at early times dominated by those seen when photoexciting the porphyrinic sites at 420 nm excitation (Figure S8). This is consistent with weak CD-centered transient absorptions, as typically observed in CD-based systems.³⁵⁻

38”

Figure S8. Differential absorption spectra of H₂P@CD (left) and CuP@CD (right) obtained from fsTAS upon 350 nm excitation in water at room temperature.

Reviewer's Point #6:

The mechanism of photo-induced electron transfer from CuP@CD to various sacrificial electron donors is not clearly explained. A diagram illustrating the energy levels of the CDs along with those of the quenchers is needed to better elucidate the electron transfer mechanism.

Reply to the Reviewer's Point #6:

We agree with reviewer to clarify the mechanism of photo-induced electron transfer in our system. To address this point, we have determined the valence band maximum of CuP@CD experimentally *via* valence band XPS (Figure S15b). The conduction band minimum has then been estimated using the optical bandgap derived from the PL emission spectrum, applying the Kasha–Vavilov rule. The latter emphasized that emission occurs from the lowest excited state and, thereby, correlates with the conduction band edge. Based on these values, we have constructed a comprehensive energy level diagram (Figure S15c) that compares the band positions of CuP@CD with the redox potentials of the sacrificial electron donors (SEDs) and methyl viologen (MV²⁺). Accordingly, electron donation from all SEDs, that is, ascorbic acid, EDTA, and TEOA, to the photoexcited CuP@CD is thermodynamically favored. Among them EDTA and TEOA stand out and this trend is in line with our experimental results (Figure S15a). It is, however, important to consider that thermodynamics alone do not impact the electron transfer efficiency. As highlighted by Etienne et al. (Comptes Rendus Chimie, 2016, 10.1016/j.crci.2015.11.026), oxidation potentials of SEDs vary widely as a function of pH, solvent, and specific molecular interactions.

With a focus on TEOA, it combines at pH 9 a sufficient driving force, a rapid electron donation kinetic, and a minimal interference with MV²⁺. Per se, what results is the highest photocatalytic performance. Please note that the electron transfer cascade in CuP@CD is only active under basic conditions. In the pH range between 8 and 9, TEOA exists predominantly in its deprotonated form, which is a stronger electron donor than the protonated form. In short, the synergy stemming from a pH-dependent catalytic activity and a favorable acid–base speciation of TEOA provides the most efficient mode of action in our photocatalytic system.

Changes in the revised versions of the manuscript and supporting information:

(pg. 10) "We take this trend to postulate a favorable band alignment, an efficient electron donation from deprotonated TEOA, and an electron transfer reactivity of photoexcited CuP@CD – all under basic conditions (Figure S15, c).⁴⁷"

Figure S15. (a) Generation of photo-reduced methyl viologen (MV^{2+}) after photo-irradiation (100 mW/cm^2 , solar light) using various sacrificial electron donors at different pH values. (b) Valence band XPS spectrum of CuP@CD. (c) Schematic illustration of the energy band structure of CuP@CD, shown together with the redox potentials of ascorbic acid (AA), ethylenediaminetetraacetic acid (EDTA), triethanolamine (TEOA), and methyl viologen (MV^{2+}).

Reviewer's Point #7:

There are several typographical errors throughout the main text and SI. The authors are advised to thoroughly check the entire document.

Reply to the Reviewer's Point #7:

We thank the reviewer for pointing this out. We have thoroughly revised the manuscript and the supporting Information to correct typographical errors and improve overall clarity.

Reviewer #2

Reviewer's Comment:

The author successfully developed the first copper monomer-enhanced carbon dot (CDs) enzyme-like photocatalyst. This material precisely mimics the electron transfer cascade of cytochrome c oxidase (Cco), achieving high selectivity in the photocatalytic reduction of oxygen to generate H₂O₂ at room temperature. Its activity surpasses that of all known metal-based catalysts. There are some interesting results; however, there still exist some problems that need to be corrected and explained. The following are some points that might be useful to refine the article.

Reply to the Reviewer's Comment:

We greatly appreciate the positive evaluation of our work and the specific comments, which has helped us to improve our manuscript.

Reviewer's Point #1:

The current title fails to fully reflect the application breakthrough of the work. It is suggested that the relevant applications be highlighted in the title.

Reply to the Reviewer's Point #1:

This is indeed a valuable suggestion. In response, we have revised the manuscript title to better reflect the photocatalytic application of our system. The new title reads: "Single-Atom Engineered Carbon Dots Mimic Cytochrome c Oxidase for Photocatalytic Oxygen Reduction." This updated title emphasizes both the materials innovation and the bioinspired catalytic function, as recommended.

Changes in the revised versions of the manuscript and supporting information:

(pg. 1) "Single-Atom Engineered Carbon Dots Mimic Cytochrome c Oxidase for Photocatalytic Oxygen Reduction."

Reviewer's Point #2:

The three distinct fluorescent CD areas respectively refer to which three? It is recommended to provide more detailed explanations in the text. No obvious single Cu atom bright spots were observed in the HAADF-STEM images.

Reply to the Reviewer's Point #2:

We thank the reviewer for pointing this out. To improve the clarity, we have revised the main text to explicitly define the three emissive regions: (i) CD-centered emission (450–515 nm), and (ii) and (iii) two porphyrinoid-centered emissions at 650 nm and 680 nm, denoted as P650 and P680. These assignments are supported by distinct spectral and lifetime characteristics (Figure 1, S2).

Regarding the HAADF-STEM imaging, we have replaced the original panel with a new version (Figure 2d), where the uniform distribution of bright atomic-scale features is clearly discernable. While direct visualization of individual Cu atoms on a light carbon background remains inherently challenging, the improved resolution supports a homogeneous dispersion of Cu within the matrix.

Changes in the revised versions of the manuscript and supporting information:

(pg. 5) "Considering the aforementioned these three distinct emissions are assigned to (i) CD-centered emission in the 450–515 nm range, (ii) porphyrinoid emission at 650 nm (P650), and (iii) porphyrinoid emission at 680 nm (P680). Not only that these species differ in terms of their emission maxima, but also in terms of their radiative decays. As such, H₂P@CD is composed of structurally distinct emissive centers."

Figure 2. a, TEM image of CuP@CD with corresponding size distribution. b, c EDS chemical mapping of CuP@CD. Panel b shows the high-resolution Cu mapping obtained from HyperMap data using the Automatic Filter mode, while panel c shows the corresponding mapping processed with the Smooth Filter (parameter = 3). The inset highlights the uniform distribution of Cu signals across CD, confirming the spatial homogeneity of Cu sites within the CuP@CD ensemble. d, HAADF-STEM image of two individual CuP@CD. e, XANES spectra of CuTCPP (light blue) and CuP@CD (brown), with an inset showing an enlarged view of the pre-edge region. f, Fourier-transformed EXAFS data of CuP@CD and its simulation, using a single nitrogen shell as nearest neighbors. g, High-resolution XPS spectra of Cu 2p for CuTCPP and CuP@CD. h, i, High-resolution XPS spectra of the N 1s and O 1s regions for H2P@CD and CuP@CD. j, FT-IR spectra of H2P@CD and CuP@CD. k, CW X-band (9.07 – 9.08 GHz, T = 80 K) EPR spectra of CuP@CD recorded at different pH values in a frozen matrix of water.

Reviewer's Point #3:

The EDS only shows the Cu mapping, lacks the comparison of C/N/O carrier elements, and the resolution is insufficient to conclude that the Cu monatomic species are uniformly distributed on the carbon dots.

Reply to the Reviewer's Point #3:

We thank the reviewer for this valuable suggestion. To address this point, we have added a detailed inset in Figure 2 (panel c) and reprocessed the Cu mapping using the Smooth Filter mode (parameter = 3) to enhance spatial clarity and facilitate a more accurate visualization. Furthermore, we would like to emphasize that the presence of metal nanoparticles or clusters has been conclusively ruled out by complementary techniques, including HAADF-STEM, EXAFS, XANES, and EPR. Taken together, these results provide strong evidence that Cu is present as monoatomic species uniformly distributed across the carbon-dot matrix. This conclusion is further supported by the elemental comparison with carrier elements (C, N, and O) shown in Figure S10.

Changes in the revised versions of the manuscript and supporting information:

(pg. 8) "Elemental mapping (EDS) showed a homogeneous distribution of Cu²⁺ across the CuP@CDs (Figure 2b, c; Figure S10)."

Figure S10. EDS chemical mapping of CuP@CD showing the spatial distribution of C, N, O, and Cu elements. Detailed insets are provided to allow direct comparison between elements and to highlight the homogeneous distribution of Cu relative to the carbon-dot matrix.

Figure 2. a, TEM image of CuP@CD with corresponding size distribution. b, c EDS chemical mapping of CuP@CD. Panel b shows the high-resolution Cu mapping obtained from HyperMap data using the Automatic Filter mode, while panel c shows the corresponding mapping processed with the Smooth Filter (parameter = 3). The inset highlights the uniform distribution of Cu signals across CD, confirming the spatial homogeneity of Cu sites within the CuP@CD ensemble. d, HAADF-STEM image of two individual CuP@CD. e, XANES spectra of CuTCCP (light blue) and CuP@CD (brown), with an inset showing an enlarged view of the pre-edge region. f, Fourier-transformed EXAFS data of CuP@CD and its simulation, using a single nitrogen shell as nearest neighbors. g, High-resolution XPS spectra of Cu 2p for CuTCCP and CuP@CD. h, i, High-resolution XPS spectra of the N 1s and O 1s regions for H2P@CD and CuP@CD. j, FT-IR spectra of H2P@CD and CuP@CD. k, CW X-band (9.07 – 9.08 GHz, T = 80 K) EPR spectra of CuP@CD recorded at different pH values in a frozen matrix of water.

Reviewer's Point #4:

The author merely emphasized the absence of the Cu^+/Cu^0 characteristic peaks, but did not provide clear evidence for the existence of Cu^{2+} .

Reply to the Reviewer's Point #4:

We believe that we have clearly demonstrated the presence of Cu^{2+} in CuP@CD, as supported by multiple, complementary spectroscopic techniques presented in both the main text and Supplementary Information. First, XPS (Figure 2g) clearly shows the characteristic Cu^{2+} $2p_{3/2}$ and $2p_{1/2}$ features along with their satellite features. This assignment is further corroborated by Cu LMM Auger spectroscopy (Figure S11), which confirms the oxidation state based on kinetic energy shifts. Second, XANES clearly indicates Cu^{2+} in CuP@CD as the XANES first derivative (Figure R2) features a first peak that is reminiscent to what is seen for CuTCCP.

For Cu^+ we would expect a peak at around 8981 eV, which is clearly absent (see for instance Phys. Chem. Chem. Phys. 2003, 10.1039/b305810g or Phys. Chem. Chem. Phys. 2019, 10.1039/c9cp06478h). Third, EXAFS results (Figures 2e–f, S12–S14, and Table S1) reveal a local coordination environment that is fully consistent with a Cu–N bonding in porphyrinoid systems and excludes contributions from Cu^0 . Fourth, EPR spectroscopy (Figure 2k) provides a distinct and unambiguous Cu^{2+} signal with *g*-values and *A*-tensor parameters in agreement with literature-reported square-planar or pseudo-pentacoordinate Cu^{2+} . Fifth, light-induced EPR shows the photoinduced reduction of Cu^{2+} to Cu^+ under basic conditions, further validating that the initial oxidation state to be Cu^{2+} (Figure 3). Taken together, these results provide clear and robust evidence for the presence and structural integration of Cu^{2+} within the CuP@CD system.

Figure R2. First derivative of the XANES spectra of CuT CPP and CuP@CD.

Reviewer's Point #5:

What is the reason for the decrease in the Cu 2p XPS binding energy of CuP@CD compared to CuT CPP? Is it due to axial coordination? It is recommended to provide a detailed explanation of the cause of this change.

Reply to the Reviewer's Point #5:

We thank the reviewer for this valuable comment. The decrease in the Cu 2p binding energy observed for CuP@CD compared to the CuT CPP benchmark is attributed to a more dynamic and flexible coordination environment within the CD matrix. In contrast to the rigid, highly conjugated porphyrin ring in CuT CPP, the porphyrinoid sites in CuP@CD experience axial interactions with pendant oxygen- and nitrogen-containing groups from the surrounding CD domains, as supported by HR-XPS (Figure 2h,i) and FTIR spectroscopy (Figure 2j). As a matter of fact, EXAFS analysis (Table S1) indicates an average coordination number that is in CuP@CD significantly larger than in CuT CPP and that prompts to an additional ligand. This is in line with a larger disorder and/or broader bond length distribution (Table S1, term σ^2). A reduced average distance, that is, 1.96 Å in CuP@CD versus 1.99 Å in CuT CPP reflects the respective distortion without, however, impacting the oxidation state – see **Reply to the Reviewer's Point #4**. Additionally, EPR analysis reveals a pH-dependent evolution from an axially symmetric Cu^{2+} environment at pH 4 to a rhombically distorted, likely penta-coordinated state at pH 9 (Figure 2k), indicating changes in the local ligand field. This more electron-rich environment weakens the ligand field around Cu^{2+} , resulting in a shift to lower binding energy in the XPS spectra. These observations are consistent with prior reports (Nanoscale 2016, 10.1039/C6NR05878G), where tautomerism and pH-induced changes in CD chemistry significantly influenced their electronic properties.

Changes in the revised versions of the manuscript and supporting information:

(pg. 8) "This coordination environment is more electron-rich when compared to the coordination in CuT CPP, which leads to a shift toward lower Cu 2p binding energies, as observed in Figure 2g. A minor component seems, however, to be identical to the coordination in CuT CPP (Figure 2g)."

Reviewer's Point #6:

How to calculate the yield of hydrogen peroxide generated in the CuP@CD photocatalytic ORR process? The authors did not clearly explain the experimental basis for calculating the H₂O₂ yield. They only relied on the TEMPO-EPR spectrum for semi-quantitative analysis. However, this method lacks a standard concentration-signal intensity calibration curve, and the linear relationship between the TEMPO[•] signal and the H₂O₂ concentration has not been verified.

Reply to the Reviewer's Point #6:

We have now included in the ESI a full, instrument-specific calibration procedure for TEMPO quantification by EPR spectroscopy. This includes preparation of standard TEMPO solutions, acquisition parameters, normalization for microwave power, and construction of a linear calibration curve ($R^2 = 0.9935$). Using this calibration, we quantify TEMPO consumption during photocatalysis and convert it to the corresponding amount of photogenerated superoxide, from which the H₂O₂ yield is calculated. The mechanistic basis is detailed in (Eqs. 1-4). Control experiments with PBN and DMPO confirm that TEMPO tracks [•]OOH formation rather than H₂O₂ decomposition, and the kinetic analysis under steady-state [•]OOH conditions follow a pseudo-second-order rate law in TEMPO. The full calculation steps, both for single-electron and radical recombination pathways are presented in the revised ESI.

Changes in the revised versions of the manuscript and supporting information:

(pg. 14) "In ambient air, H₂O₂ production rate is 7.9 mmol·g⁻¹·mL⁻¹·h⁻¹, as determined from kinetic analyses – details are gathered in the Supporting Information."

H₂O₂ photoproduction calculation

Under photoexcitation, the O₂ molecules bound to the copper-containing active site (CuP) in the CuP@CD photocatalyst act as acceptors for the photogenerated e⁻ delivered from the carbon dot domain (CD). A series of O₂ reduction processes take place, which are highlighted by Eq. 1 to Eq. 4. Spin-active intermediates are formed during the O₂ reduction that can be trapped by spin-probes and analysed by EPR technique. We observed that the presence of [•]OH radicals that may form from the photocatalytic H₂O₂ breakdown (see Eq. 5-6) under light irradiation (@325 nm) were excluded by both N-tert-Butyl- α -phenylnitrone (PBN) and 5,5-Dimethyl-1-pyrroline N-oxide (DMPO) spin-trap experiments. This evidence rules out a four-electron reduction pathway or a Fenton-like/electron-driven H₂O₂ decomposition. Therefore, formation of H₂O₂ as final product of the ORR catalyzed by CuP@CD occurs through one electron reduction pathway of O₂, which produces superoxide radical anions (Eq. 1, Eq. 2 and Eq. 3). These represent key steps in the ORR.¹

To elucidate the evolution of superoxide radicals in solution, we employed the TEMPO radical ([•]T) as a spin-probe and monitored the process of its oxidation by electron paramagnetic resonance (EPR) spectroscopy. The concentration of the superoxide intermediate species serves as an indicator of peroxide formation. The temporal decay of the TEMPO radical was recorded under 325 nm irradiation in an oxygen-saturated aqueous solution containing a hole scavenger and the photocatalyst. In the oxygen reduction reaction (ORR) pathway, the superoxide radical anion (O₂^{•-}) generated via CuP@CD photocatalysis functions as a one-electron oxidant toward TEMPO, oxidizing TEMPO ([•]N–O[•]) to the corresponding oxoammonium cation (TEMPO⁺, =N⁺–O⁻).

The decay of the TEMPO signal at a fixed initial concentration (0.195 mM) was quantified by double integration of the EPR spectra, with intensities normalized to the square root of the applied microwave power (P). Measurements were first recorded under dark conditions ($t = 0$ s) and subsequently during irradiation as a function of time. At each time point, the integrated signal intensity ($\int \text{EPR Int.}/\sqrt{P}$) was converted to TEMPO concentration using a pre-established calibration curve (presented in the following section). The decay profile exhibited second-order kinetic in TEMPO ($[\cdot\text{T}]$), rather than first-order behaviour. This result is consistent with a mechanism in which O_2 activation proceeds *via* binding to the CuP active center. However, both TEMPO concentration and superoxide radical anion concentration are variables that dictates how fast the TEMPO disappearance occurs in solution. In support of this observation, purging the aqueous solution with N_2 , thereby reducing considerably the dissolved O_2 concentration, led to a decrease in the TEMPO decay rate (see Figure 4f main manuscript text).

$$\text{TEMPO disappearance rate} = k_{\text{obs}} [\cdot\text{T}]^2 [\text{O}_2^{\cdot-}]$$

Under steady-state conditions for the superoxide radical anion and with the $\text{TEMPO} + \text{O}_2^{\cdot-}$ step as rate-limiting, the reaction follows a pseudo-second-order rate law in TEMPO: $-d[\cdot\text{T}]/dt = k [\cdot\text{T}]^2$, being $k = k_{\text{obs}} [\text{O}_2^{\cdot-}] = \text{const}$:

$$\frac{1}{[\cdot\text{T}]_t} - \frac{1}{[\cdot\text{T}]_0} = k \cdot t$$

Where $[\cdot\text{T}]_t$ is the concentration of TEMPO radical at the time t under @325 nm irradiation, $[\cdot\text{T}]_0$ is the concentration of TEMPO (fixed, $0.195 \text{ mM} = 1.95 \times 10^{-4} \text{ M}$) recorded initially ($t = 0$) under dark conditions and k is the second order rate constant ($\text{M}^{-1}\text{s}^{-1}$). By plotting $[\cdot\text{T}]_t$ vs t (s) the linear fitting gives the k value, as angular coefficient (see Figure 4f, main manuscript text). Thus:

$$\text{Rate constant } k = 48.56 \text{ M}^{-1} \text{ s}^{-1}$$

We can then calculate the amount of TEMPO radical left at the time t , being $t = 3600 \text{ s}$ (@325 nm)

$$[\cdot\text{T}] = \frac{1}{\frac{1}{[\cdot\text{T}]_0} + kt} = \frac{1}{\frac{1}{1.95 \times 10^{-4}} + 48.56 \cdot 3600} = 5.55728 \times 10^{-6} \text{ M} \quad \text{Eq. 7}$$

The TEMPO consumed in 1 h = $[\cdot\text{T}]_0 - [\cdot\text{T}]_t = 1.95 \times 10^{-4} - 5.6 \times 10^{-6} = 1.894 \times 10^{-4} \text{ M}$, translates with the TEMPO radical that reacted (forming oxoammonium cation) with the photogenerated superoxide radicals at CuP site (Eq. 1-2 right) $1.894 \times 10^{-4} \text{ M}$ of $\cdot\text{OOH}/\text{O}_2^{\cdot-} = 0.0001894 \text{ mmol/mL}$.

The amount of photoproduced H_2O_2 is finally calculated as follows:

Assuming indirect two-steps, single-electron reduction ORR pathway (see Eq. 1 and Eq. 4):

$$\cdot\text{OOH}/\text{O}_2^{\cdot-} = (0.0001894 \text{ mmol/mL h}) \cdot 0.120 \text{ mL} = 0.0000227331 \text{ mmol/h}$$

$$\text{Amount of catalyst} = 0.8 \text{ mg/mL taken } 0.03 \text{ mL. Mass catalyst} = 0.024 \text{ mg} = 0.000024 \text{ g}$$

$$(0.0000227331 \text{ mmol/h}) / 0.000024 \text{ g} = 0.94721 \text{ mmol/g h}$$

$$\text{Divided by the reaction volume } (0.94721 \text{ mmol/g h}) / (0.120 \text{ ml}) = \mathbf{7.89 \text{ mmol g}^{-1} \text{ mL}^{-1} \text{ h}^{-1}}$$

Assuming parallel radical recombination pathway (see Eq. 3):

$$\frac{1}{2} \cdot \text{OOH} = (0.0000947 \text{ mmol/mL h}) \cdot 0.120 \text{ mL} = 0.000011364 \text{ mmol/h}$$

Amount of catalyst = 0.8 mg/mL taken 0.03 mL. Mass catalyst = 0.024 mg = 0.000024 g

(0.000011364 mmol/h) / 0.000024 g = 0.4735 mmol/g h

Divided by volume (0.94721 mmol/g h) / (0.120 ml) = **3.95 mmol g⁻¹ mL⁻¹ h⁻¹**

Experimental procedure for generating the X-band EPR signal intensity calibration curve using the TEMPO radical as a spin concentration standard

A total of 9.9 mg of the TEMPO radical (2,2,6,6-Tetramethylpiperidine 1-oxyl, C₉H₁₈NO, 98% purity, CAS No. 2564-83-2; Sigma-Aldrich) was thoroughly dispersed in 10 mL of deionized water (CAS No. 7732-18-5; Sigma-Aldrich, Ultrapur) to prepare a stock solution with high TEMPO concentration ([TEMPO] = 6.34 mM). This stock solution was stored frozen (-18 °C) in dark, where it remained stable for months without radical degradation. A series of freshly prepared TEMPO solutions with varying concentrations were then obtained by serial dilution with deionized water of the above stock solution (*vide infra*). These dilutions were used to generate a calibration curve that correlate the EPR signal intensities with diverse TEMPO concentrations (from 0.0024 mM to 0.300 mM) and to define the EPR signal-detection range under specific sets of acquisition parameters used throughout the CuP@CD study.

Table S2. From the highly concentrated stock solution of TEMPO in water (6.34 mM), 0.1 mL were taken and diluted to 1.0 mL (final Volume) by adding deionized water (added 0.9 mL of H₂O). This solution is coded in the Table as Solution (A) = 0.634 mM

Target [TEMPO] (mM)	Stock Volume (V ₁ , μL) Solution A	Water Volume (μL)	Notes
0.3000	473.2	526.8	≈ 473 μL stock A + 527 μL water
0.2550	402.2	597.8	≈ 402 μL stock A + 598 μL water
0.1940	306.0	694.0	306 μL stock A + 694 μL water
0.1500	236.6	763.4	≈ 237 μL stock A + 763 μL water
0.0970	153.0	847.0	153 μL stock A + 847 μL water (Solution B)
0.0485	500.0 (from Solution B)	500.0	500 μL stock B + 500 μL water (Solution C)
0.0243	501.0 (from Solution C)	499.0	501 μL stock C + 499 μL water (Solution D)
0.0024	98.8 (from Solution D)	901.2	≈ 100 μL stock D + 900 μL water

Due to the high dielectric constant of water ($\epsilon = 78.4$ at 298 K), which causes substantial microwave absorption, poor resonator coupling, and reduced sensitivity, particularly at room temperature, coupled with the water shallow microwave penetration depth, each TEMPO solution was loaded into WHEATON[®] capillary tubes (CAS number: 851321, 1-5 μL; Sigma-Aldrich). The capillaries featured a dark marked-scale grading, to ease consistency of the sample loading throughout the series. The loaded capillaries were finally inserted into 0.5 OD Wilmad[®] Suprasil EPR tubes (730-SQ-250M), which were then sealed using Precision Seal[®] rubber septa. In this way, the inner EPR tube atmosphere can further be controlled when needed, such as by saturating the local environment with N₂. Figure S25 shows, practically, the experimental set-up. X-band EPR measurements were all conducted at $T = 293$ K, using identical acquisition parameters across the TEMPO samples used in the calibration-curve settings: 100 kHz modulation frequency, 30 ms time constant, 8 × 100 gain, 0.3 mT modulation width, 0.500 mW microwave power, one scan and 1 minute of signal-acquisition time. The quality factor (Q) of the X-band EPR resonator cavity remained within the range 6100 - 6500 across the tested solutions under X-band microwave irradiation (for the empty X-band resonator cavity $Q \cong 8800$)

and optimal loading of 1 μL . In fact, for obtaining good signal-intensity vs concentration calibration curve, all these experimental variables must be optimized, because the detected EPR signal voltage (V) depends on:

$$V_{EPR} = K \cdot \sqrt{P_0} \cdot Q \cdot \eta \cdot \chi'' \cdot G \quad (\text{Eq. 8})$$

It is convenient to divide the EPR signal voltage (V), namely the EPR intensity, for the square root of the applied microwave power (expressed in microwatt, μW) such that:

$$V_{EPR} / \sqrt{P_0} = K \cdot Q \cdot \eta \cdot \chi'' \cdot G \quad (\text{Eq. 9})$$

K = detector sensitivity, $\sqrt{P_0}$ = square root of applied microwave power (must be far from saturating conditions), Q = cavity quality factor, η = filling factor (dimensionless, represents the fraction of microwave magnetic field interacting with the sample), χ'' = imaginary part of the RF susceptibility, G = gain of the detection system. The resulting EPR spectra ($d\chi''/dB$, derivative of the magnetic susceptibility with respect to field) obtained by varying TEMPO concentration are shown in Figure S26. These resonance signals were divided for the square root of microwave power, as shown in Eq. 9, and then double integrated (D.I.). The resulting values (D.I.) were subsequently plotted against the known TEMPO concentrations (mM) to generate the linear calibration curve (y (D.I.) = $a + b \cdot x$ (mM)), with coefficients $a = 16.2$ and $b = 11963.1$ shown Figure S27.

Figure S25. Experimental set-up for the TEMPO radical measurements performed with X-band EPR spectroscopy as used throughout the study, including the determination of the TEMPO EPR signal calibration curve.

Figure S26. X-band (9.08 GHz) TEMPO radical EPR signal recorded for various TEMPO concentrations.

Figure S27. Liner plot of the double-integrated EPR intensities (circles) of the TEMPO resonance signals vs concentration of TEMPO (mM). The solid line (red) shows the linear fitting, with coefficient values (a , b) given in the plot.

Reviewer's Point #7:

What is the main intrinsic reason why CuP@CD outperforms other photocatalysts? How does the specific site electron transfer of CcO promote efficient conversion.

Reply to the Reviewer's Point #7:

We are thankful for the opportunity to respond. The superior performance of CuP@CD is attributed to the rational integration of nature-inspired design principles with molecular-level control over redox site positioning. The result is a unique combination of site-specific electron transfer, efficient charge separation, and optimized catalytic cycling.

At the heart of this design is the bioinspired spatial separation of function: CDs serve as light absorbers and photogenerated electron reservoirs, while the porphyrinoid domains, selectively metallated with Cu²⁺, act as redox-active catalytic centers, analogous to the CuB site of CcO. Such a spatial decoupling emulates the electron cascade in natural enzymes, where directional charge flow and redox tuning across cofactors enable high selectivity and turnover. This behavior is supported by a broad set of spectroscopic techniques.

The importance of structural anisotropy and site resolution was demonstrated in our earlier work (Small 2023, 10.1002/sml.202206587), where we showed that spatial asymmetry in CD-based architectures promotes the stabilization of long-lived, charge-separated states, facilitating productive photoredox pathways. In related studies (JACS 2021, 10.1021/jacs.1c07049; Angew. Chem. 2025, 10.1002/anie.202418626), we highlighted fluorophore-like molecular subunits as the main catalytically active sites in CDs. Building on these concepts, CuP@CD introduces single-atom Cu²⁺ centers into porphyrinoid scaffolds, adding an additional layer of redox selectivity and spatial precision, and bringing the system closer to the efficiency and modularity of natural metalloenzymes. Together, these features enable a well-organized, enzyme-mimicking photocatalyst that benefits from directional charge transport, minimized recombination, and site-specific reactivity at embedded Cu²⁺ centers. This distinguishes CuP@CD from systems with randomly deposited metal sites or unstructured architectures, where light absorption and catalysis are often less efficiently coordinated.

Changes in the revised versions of the manuscript and supporting information:

(pg. 15) "Overall, these results highlight the functional synergy between CDs as electron reservoirs and Cu²⁺ as catalytic sites to set up an enzyme-like spatial and redox organization. To this end, an electron transfer upon photoexcitation occurs exclusively at the CuP sites, which leaves the function of the CDs to act as a light-harvester and electron-donor. By combining spatial compartmentalization with functional differentiation, a highly directional electron transfer cascade is established. This is very similar to what is known for CcO, where individual cofactors facilitate a series of redox reactions. A biomimetic arrangement like that in CuP@CD accelerates charge separation, suppresses charge recombination, and fosters high H₂O₂ production rates. It thereby distinguishes it from more traditional photocatalytic systems that lack structural order."

Reviewer's Point #8:

What is the conversion rate of H₂O₂ produced by CuP@CD in photocatalysis, what is the apparent quantum yield (AQY), and how is the cycle stability.

Reply to the Reviewer's Point #8:

We appreciate raising the important question regarding the photocatalytic conversion rate, apparent quantum yield (AQY), and stability of the CuP@CD catalyst. These parameters are indeed critical for evaluating the applicability of any photocatalytic system, and we appreciate the opportunity to clarify them in greater detail (Please see Supporting Information pg. 31, 32). The O₂-to-H₂O₂ conversion efficiency was determined using quantitative LEPR experiments with the well-established TEMPO spin-trap protocol. Based on the solubility of O₂ in water at 25 °C (8.5 mg/L, corresponding to 0.266 mmol/L), the amount of dissolved O₂ in our 120 μL reaction volume was calculated to be 31.9 nmol. Upon photoirradiation, the CuP@CD

catalyst produced 22.7 nmol of $\cdot\text{OOH}$ within one hour, translating to an O_2 -to- H_2O_2 conversion efficiency of approximately 71.2% *via* the indirect two-step, single-electron reduction pathway. For the parallel radical recombination pathway, the conversion efficiency was estimated to be 35.6%. These results confirm efficient utilization of the available oxygen pool under ambient conditions, underscoring the high activity and selectivity of the CuP@CD catalyst.

The AQY was calculated following the methodology outlined in Nat. Commun. 2021, 10.1038/s41467-020-20823-8, based on the number of photoinduced electrons involved in the H_2O_2 formation relative to the total number of incident photons. Under 325 nm photoexcitation at a power density of 40 mW cm^{-2} , the AQY was determined to be 1.48%, which is comparable with previous reports (Nat. Commun. 2021, 10.1038/s41467-020-20823-8). Taken together, these results highlight the relevance of CuP@CD for applications in artificial photosynthesis.

We assessed the photocatalytic cycle stability of CuP@CD by tracking the photo-charges with in-situ light-off/on EPR at X-band (9.708 GHz) under 325 nm illumination in water (pH= 9, with 0.1 M of TEOA), using the EPR tube loaded with 120 μL of solution as reaction vessel. We decreased the concentration of the CuP@CD catalyst by a factor 5 ($0.024/5 = 0.0048 \text{ mg}$ of CuP@CD) to better monitor the $g \sim 2.00$ region, where e^-/h^+ arise from CuP@CD photoexcitation, in a way to minimize the strong resonance tail (g_{\perp}) coming from the Cu ($S = 1/2$) cation. In a frozen glass at 80 K, no signal is detected in the selected B-region in the dark. Upon illumination, two features appear due to an electron polaron on CuP@CD ($g_e \approx 2.000$) and a TEOA-derived α -aminoalkyl radical (giving characteristic ^{14}N and two β - ^1H hyperfine couplings, $g_{\text{eff}} \approx 2.003$, estimated ^{14}N hyperfine of 2.0 mT, estimated ^1H hyperfine 2.3 mT) arising from the scavenging of the photoexcited h^+ by TEOA. When light is switched off, both signals decay to the original baseline. Figure S31 shows the in-situ EPR 2D plot highlighting the formation of resonance signals (X-axis, magnetic field) upon photoirradiation (325 nm, light ON) versus time (Y-axis, $t = \text{min}$) during the off to UV-on cycles. The color scale tracks the variation in EPR intensity (dc''/dB). We found reproducibility throughout two consecutive cycles: the integrated signal intensity under light-on in cycle 2 equals $98 \pm 2 \%$ of cycle 1, and the residual intensity after lights-off is $\ll 1 \%$ (see EPR spectra in the right panels, corresponding to resonances recorded under specific time frames). Controls without TEOA show only the electron-related signals. Controls without CuP@CD and/or without light show no features at all. This proves the hole-scavenging as the origin of the TEOA radical. This phenomenon is consistent with reversible charge generation in CuP@CD and the absence of spin-defect accumulation in the photocatalyst. Together, these data demonstrate that CuP@CD sustains repeated charge separation with full reversibility and preserves catalytic performance under cycling.

Changes in the revised versions of the manuscript and supporting information:

(pg. 15) "Notably, the fact that 71.2% of dissolved O_2 was converted to H_2O_2 within one hour – based on a two-step, single-electron reduction ORR pathway – confirms an effective consumption of O_2 . The apparent quantum yield (AQY), calculated based on a photon-to-electron conversion at 325 nm excitation, was 1.5%."

(pg. 15) "Stability tests with CuP@CD under photocatalytic conditions were ran by means of XPS and EPR measurements (Figure S30, S31)."

Figure S31. (a) 2D heat map generated from X-band (9.70795 GHz) measurements of CuP@CD in water (pH 9) in the presence of TEOA (0.1M) at 80 K under light on ($\lambda_{\text{ex}} = 325$ nm) and light off conditions. The 2D plot shows formation of photoexcited charges (electrons/holes) and their disappearance when light is off to highlight the reversibility of the photoexcitation. (b) Different time frames during the first and second cycle under light on ($\lambda_{\text{ex}} = 325$ nm) and light off conditions.

Conversion rate of H₂O₂ produced ORR

For the dissolved oxygen content, the solubility of O₂ in water at 25 °C is reported as 8.5 mg/l. Converting this to molar concentration:

$$[\text{O}_2] = (8.5 \text{ mg/l}) / (32 \text{ g/mol}) = 0.266 \text{ mmol/l}$$

$$\text{In } 120 \text{ }\mu\text{l: } 0.266 \text{ mmol/l} \cdot 0.00012 \text{ l} = 31.9 \text{ nmol}$$

Therefore, the O₂-to-H₂O₂ conversion efficiency is:

Indirect two-step, single-electron reduction ORR pathway:

$$\text{Conversion} = (22.7 \text{ nmol} / 31.9 \text{ nmol}) \cdot 100 = \mathbf{71.2\%}$$

Parallel radical recombination pathway:

$$\text{Conversion} = (11.35 \text{ nmol} / 31.9 \text{ nmol}) \cdot 100 = \mathbf{35.6\%}$$

Apparent Quantum Yield (AQY) calculation

AQY was calculated following the methodology described in reference,² specifically using the number of photo-induced electrons involved in H₂O₂ formation relative to the total number of incident photons.

$$\text{Irradiated area: } A = \pi r^2 = \pi \cdot (0.05)^2 = 0.007854 \text{ cm}^2$$

$$\text{Laser power: } P = 40 \text{ mW/cm}^2 \cdot 0.007854 \text{ cm}^2 = 0.31416 \text{ mW} = 0.00031416 \text{ J/s}$$

$$\text{Photon energy at 325 nm: } E_{\text{photon}} = (6.626 \times 10^{-34} \cdot 3 \times 10^8) / 325 \times 10^{-9} = 6.12 \times 10^{-19} \text{ J}$$

$$\text{Total photons: } N_{\text{photons}} = (1.131 \text{ J}) / (6.12 \times 10^{-19} \text{ J/photon}) = 1.85 \times 10^{18}$$

$$\text{Number of e}^- \text{ involved in reaction: } 0.00002273 \text{ mmol H}_2\text{O}_2 / h = 2.273 \times 10^{-8} \text{ mol}$$

$$\text{Since } 2 \text{ e}^- \text{ per H}_2\text{O}_2, \rightarrow \text{total electrons} = 4.546 \times 10^{-8} \text{ mol}$$

$$N_e = 4.546 \times 10^{-8} \cdot 6.022 \times 10^{23} = 2.74 \times 10^{16} \text{ electrons}$$

$$A_{\text{QY}} = (1.85 \times 10^{18} / 2.74 \times 10^{16}) \cdot 100 = 1.48\%$$

Reviewer's Point #9:

The experimental details of the light-driven ORR for the production of H₂O₂ need to be provided in greater detail.

Reply to the Reviewer's Point #9:

We thank the reviewer for this important observation. We fully acknowledge that the experimental details provided in the original version were insufficient for full reproducibility. We sincerely apologize for this oversight. In response, we have revised the manuscript to include comprehensive information regarding the photocatalytic H₂O₂ production protocol. This includes specifications of the light source, reaction setup, catalyst loading, reaction medium composition, and the LEPR measurement conditions.

Changes in the revised versions of the manuscript and supporting information:

(pg. 22) "Photocatalytic H₂O₂ production experiments were carried out using a xenon lamp equipped with an AM1.5G filter, simulating one sun intensity (100 mW cm⁻²) as the irradiation source. A colloidal suspension of CuP@CD (0.5 mg) was dispersed in a total volume of 6 mL of water containing TEOA (pH 9) as the sacrificial electron donor. Light-driven ORR for H₂O₂ generation was monitored *via* LEPR spectroscopy (CW X-band, 9.07–9.08 GHz, T = 293 K) under photo-excitation at 325 nm. For these measurements, CuP@CD (0.024 mg)

was suspended in 120 μL of aqueous TEOA solution (0.1 M, pH 9) containing TEMPO (1.95×10^{-4} M). For further details see Supporting Information."

Reviewer's Point #10:

How does the CuP@CD photocatalyst perform in the actual production of H_2O_2 through photocatalysis? Why not adopt other methods to quantify H_2O_2 ?

Reply to the Reviewer's Point #10:

We are quite thankful for getting this important feedback. As also emphasized by Freese et al. (Nat. Catal. 2023, 10.1038/s41929-023-00980-x), reliable and interference-free quantification of H_2O_2 remains one of the central challenges in photocatalysis, particularly in complex reaction environments. A variety of titrimetric, colorimetric, and spectrophotometric methods are widely used. Each comes with limitations depending on the specific photocatalyst, sacrificial donor, and optical properties of the system.

In our study, we systematically tested several commonly recommended methods for H_2O_2 detection. As mentioned in the original manuscript " H_2O_2 quantification was attempted using commercial detection kits (Spectroquant H_2O_2 Test and Peroxide Assay Kit–Sigma-Aldrich). False-positive signals due to TEOA interference precluded, however, any reliable interpretation. Potassium iodide-based spectroscopic detection confirmed, on one hand, H_2O_2 production under solar irradiation but, on the other hand, lacked specificity due to interference with TEOA (Figure S22). Titrimetric methods proved also to be inconclusive due to a green coloration, which interfered with an accurate endpoint determination." In the following, we now provide additional information:

Colorimetric test kits:

We tested both the Spectroquant H_2O_2 Test ($\text{Cu}^{2+}/\text{Cu}^+$ reduction) and the Hydrogen Peroxide Test Kit–Sigma Aldrich ($\text{Fe}^{2+}/\text{Fe}^{3+}$ oxidation). Both were, however, compromised by (i) strong false-positive signals due to TEOA interference and/or (ii) significant baseline errors caused by an overlap between the catalyst's intrinsic absorption and the detection wavelengths; 450 vs. 420 nm for the Soret-band; 585 vs. 578 nm for the Q-band. These overlaps rendered a reliable baseline correction difficult in colored or turbid suspensions, potentially leading to overestimated H_2O_2 concentrations.

Titrimetric methods:

We attempted both thiosulfate titration (HYP-1 kit; starch–iodine indicator) and an additional permanganate-based assay. Both failed due to the strong green color of CuP@CD/TEOA suspensions, which obscured the the endpoint detection. Additionally, the small size of the CDs (2–3 nm, TEM) prevents their removal by filtration or centrifugation.

Iodometric (KI-based) detection:

Although widely regarded as the gold standard, this method also proved unsuitable. TEOA readily reacts with triiodide (I_3^-), compromising specificity. As shown in Figure S22a, the presence of TEOA significantly alters the I_3^- absorption profile. Furthermore, point (ii) from "Colorimetric test kits" chapter raises further concerns. While we derived a rough estimate of $\sim 2000 \mu\text{mol}\cdot\text{g}^{-1}\cdot\text{mL}^{-1}\cdot\text{h}^{-1}$ based on a comparison with a 2 mM H_2O_2 standard (Figure S22a, b), we emphasize that this value is only an approximate due to the limitations noted above.

Amperometric sensors:

We also consulted with Prominent regarding the use of DULCOTEST® hydrogen peroxide sensors. After evaluating our photocatalytic conditions, including the presence of TEOA and high oxidative potential, their technical team advised against using this method. According to their correspondence "The high oxidative

pulse and measurement potential may lead to the formation of nitrogen oxides, resulting in a false-positive signal, even at H₂O₂ concentrations as low as 0.1 ppm. Based on this, we do not recommend this measurement and will not proceed with preparing an offer."...

Taken together, these findings highlight that none of the standard detection techniques were compatible with our catalyst and reaction conditions. Therefore, we employed light-induced EPR (LEPR) using TEMPO as a selective spin probe. This method is unaffected by any optical artifacts or TEOA and is supported by spin-trap controls (PBN, DMPO) that exclude •OH interference. It thus provides a reliable and selective readout of •OOH formation, from which H₂O₂ production kinetics were derived. We trust that this explanation clarifies our rationale, and we emphasize that our goal was always to apply the most accurate and fair methodology possible given the challenges of the system.